# SOSP: Efficiently Capturing Global Correlations by Second-Order Structured Pruning

**Manuel Nonnenmacher**[*]
Bosch Center for Artificial Intelligence (BCAI)
Robert Bosch GmbH
71272 Renningen, Germany

**Thomas Pfeil**
Bosch Center for Artificial Intelligence (BCAI)
Robert Bosch GmbH
71272 Renningen, Germany

**Ingo Steinwart**
Institute for Stochastics and Applications
University of Stuttgart
70569 Stuttgart, Germany

**David Reeb**
Bosch Center for Artificial Intelligence (BCAI)
Robert Bosch GmbH
71272 Renningen, Germany

## ABSTRACT

Pruning neural networks reduces inference time and memory costs. On standard hardware, these benefits will be especially prominent if coarse-grained structures, like feature maps, are pruned. We devise two novel saliency-based methods for second-order structured pruning (SOSP) which include correlations among all structures and layers. Our main method SOSP-H employs an innovative second-order approximation, which enables saliency evaluations by fast Hessian-vector products. SOSP-H thereby scales like a first-order method despite taking into account the full Hessian. We validate SOSP-H by comparing it to our second method SOSP-I that uses a well-established Hessian approximation, and to numerous state-of-the-art methods. While SOSP-H performs on par or better in terms of accuracy, it has clear advantages in terms of scalability and efficiency. This allowed us to scale SOSP-H to large-scale vision tasks, even though it captures correlations across all layers of the network. To underscore the global nature of our pruning methods, we evaluate their performance not only by removing structures from a pretrained network, but also by detecting architectural bottlenecks. We show that our algorithms allow to systematically reveal architectural bottlenecks, which we then widen to further increase the accuracy of the networks.

## 1 INTRODUCTION

Deep neural networks have consistently grown in size over the last years with increasing performance. However, this increase in size leads to slower inference, higher computational requirements and higher cost. To reduce the size of the networks without affecting their performance, a large number of pruning algorithms have been proposed (e.g., LeCun et al., 1990; Hassibi et al., 1993; Reed, 1993; Han et al., 2015; Blalock et al., 2020). Pruning can either be *unstructured*, i.e. removing individual weights, or *structured*, i.e. removing entire substructures like nodes or channels. Single-shot pruning methods, as investigated in this work, usually consist of three steps: 1) training, 2) pruning, 3) another training step often referred to as *fine-tuning*.

Unstructured pruning can significantly reduce the number of parameters of a neural network with only little loss in the accuracy, but the resulting networks often show only a marginal improvement in training and inference time, unless specialized hardware is used (He et al., 2017). In contrast, structured pruning can directly reduce inference time and even training time when applied at initialization (Lee et al., 2018). To exploit these advantages, in this work, we focus on structured pruning.

Global pruning removes structure by structure from all available structures of a network until a predefined percentage of pruned structures is reached. Recent examples for global structured pruning methods are NN Slimming (Liu et al., 2017), C-OBD and EigenDamage (Wang et al., 2019a). Local

---

[*]Email: `manuel.nonnenmacher@de.bosch.com`

pruning, on the other hand, first subdivides all global structures into subsets (e.g. layers) and removes a percentage of structures of each subset. Recent examples for local pruning methods are HRank (Lin et al., 2019), CCP (Peng et al., 2019), FPGM (He et al., 2019) and Variational Pruning (Zhao et al., 2019). Most local pruning schemes use a predefined layer-wise pruning ratio, which fixes the percentage of structures removed per layer. While this prevents the layers from collapsing, it also reduces some of the degrees of freedom, since some layers may be less important than others. Other local pruning methods like AMC (He et al., 2018) learn the layer-wise pruning ratios in a first step.

A key challenge for global saliency-based pruning is to find an objective which can be efficiently calculated to make the approach scalable to large-scale, modern neural networks. While second-order pruning methods are usually more accurate than first-order methods (Molchanov et al., 2019), calculating the full second-order saliency objective is intractable for modern neural networks. Therefore, most saliency-based pruning methods such as OBD (e.g., LeCun et al., 1990) or C-OBD (Wang et al., 2019a) evaluate the effect of removing a single weight or structure on the loss of the neural network in isolation. However, this neglects possible correlations between different structures, which are captured by the off-diagonal second-order terms, and hence, can significantly harm the estimation of the sensitivities. Finding a global second-order method that both considers off-diagonal terms and scales to modern neural networks is still an unsolved research question.

Our main goal in this work is to devise a simple and efficient second-order pruning method that considers all global correlations for structured sensitivity pruning. In addition, we want to highlight the benefits that such methods may have over other structured global and local pruning schemes.

Our contributions are as follows:

- We develop two novel saliency-based pruning methods for second-order structured pruning (SOSP) and analyze them theoretically. We show that both of our methods drastically improve on the complexity of a naive second-order approach, which is usually intractable for modern neural networks. Further, we show that our SOSP-H method, which is based on fast Hessian-vector products, has the same low complexity as first-order methods, while taking the full Hessian into account.

- We compare the performance and the scaling of SOSP-H to that of SOSP-I, which is based on the well-known Gauss-Newton approximation. While both methods perform on par, SOSP-H shows better scaling. We then benchmark our SOSP methods against a variety of state-of-the-art pruning methods and show that they achieve comparable or better results at lower computational costs for pruning.

- We exploit the structure of the pruning masks found by our SOSP methods to widen architectural bottlenecks, which further improves the performance of the pruned networks. We diagnose layers with disproportionally low pruning ratios as architectural bottlenecks.

Related work is discussed in the light of our results in the Discussion section (Sec. 4). PyTorch code implementing our method is provided at https://github.com/boschresearch/sosp.

## 2 SOSP: SECOND-ORDER STRUCTURED PRUNING

A neural network (NN) maps an input $x \in \mathbb{R}^d$ to an output $f_\theta(x) \in \mathbb{R}^D$, where $\theta \in \mathbb{R}^P$ are its $P$ parameters. NN training proceeds, after random initialization $\theta = \theta_0$ of the weights, by mini-batch stochastic gradient descent on the empirical loss $\mathcal{L}(\theta) := \frac{1}{N} \sum_{n=1}^{N} \ell\left(f_\theta(x_n), y_n\right)$, given the training dataset $\{(x_1, y_1), \ldots, (x_N, y_N)\}$. In the classification case, $y \in \{1, \ldots, D\}$ is a discrete ground-truth label and $\ell(f_\theta(x), y) := -\log \sigma\left(f_\theta(x)\right)_y$ the cross-entropy loss, with $\sigma : \mathbb{R}^D \to \mathbb{R}^D$ the softmax-function. For regression, $y \in \mathbb{R}^D$ and $\ell(f_\theta(x), y) = \frac{1}{2} \|f_\theta(x) - y\|^2$ is the squared loss.

Structured pruning aims to remove weights or rather entire structures from a NN $f_\theta$ with parameters $\theta$. A structure can be a filter (channel) in a convolutional layer, a neuron in a fully-connected layer, or an entire layer in a parallel architecture. We assume the NN in question has been segmented into $S$ structures $s = 1, \ldots, S$, which can potentially be pruned. We define the notation $\theta_s \in \mathbb{R}^P$ as the vector whose only nonzero components are those weights from $\theta$ that belong to structure $s$.[1] Then, a

---

[1] We require that each weight is assigned to at most one structure. In practice, we associate with each structure those weights that go *into* the structure, rather than those that leave it.

*pruning mask* is a set $M = \{s_1, \ldots, s_m\}$ of structures, and applying a mask $M$ to a NN $f_\theta$ means to consider the NN with parameter vector $\theta_{\setminus M} := \theta - \sum_{s \in M} \theta_s$.

We now develop our pruning methods that incorporate global correlations into their saliency assessment by efficiently including the second-order loss terms. Our method SOSP-I allows a direct interpretation in terms of individual loss sensitivities, while our main method SOSP-H remains very efficient for large-scale networks due to its Hessian-vector product approximation.

The basic idea behind both our pruning methods is to select the pruning mask $M$ so as to (approximately) minimize the *joint* effect on the network loss

$$\lambda(M) := \big| \mathcal{L}(\theta) - \mathcal{L}(\theta_{\setminus M}) \big|$$

of removing all structures in $M$, subject to a constraint on the overall pruning ratio. To circumvent this exponentially large search space, we approximate the loss up to second order, so that

$$\lambda_2(M) = \left| \sum_{s \in M} \theta_s^T \frac{d\mathcal{L}(\theta)}{d\theta} - \frac{1}{2} \sum_{s, s' \in M} \theta_s^T \frac{d^2 \mathcal{L}(\theta)}{d\theta \, d\theta^T} \theta_{s'} \right| \tag{1}$$

collapses to single-structure contributions plus pairwise correlations.

The first-order terms $\lambda_1(s) := \theta_s \cdot d\mathcal{L}(\theta)/d\theta$ in (1) are efficient to evaluate by computing the gradient $d\mathcal{L}(\theta)/d\theta \in \mathbb{R}^P$ once and then a (sparse) dot product for every $s$. In contrast, the network Hessian $H(\theta) := d^2 \mathcal{L}(\theta)/d\theta^2 \in \mathbb{R}^{P \times P}$ in (1) is prohibitively expensive to compute or store in full. We therefore propose two different schemes to efficiently overcome this obstacle. We name the full methods SOSP-I (individual sensitivities) and SOSP-H (Hessian-vector product).

### 2.1 SOSP-I: SALIENCY FROM INDIVIDUAL SENSITIVITIES

SOSP-I approximates each individual term $\theta_s^T H(\theta) \theta_{s'}$ in (1) efficiently, as we will show in Eq. (5). We will therefore consider an upper bound to Eq. (1) which measures all sensitivities individually:

$$\lambda_2^I(M) = \sum_{s \in M} \left| \theta_s^T \frac{d\mathcal{L}(\theta)}{d\theta} \right| + \frac{1}{2} \sum_{s, s' \in M} \left| \theta_s^T H(\theta) \theta_{s'} \right|. \tag{2}$$

The absolute values are to prevent cancellations among the individual sensitivities of the network loss to the removal of structures, i.e. the derivatives $\lambda_1(s)$, and individual correlations $\theta_s^T H(\theta) \theta_{s'}$. While objectives other than $\lambda_2^I$ are equally possible in the method, including $\lambda_2$ and variants with the absolute values not pulled in all the way, we found $\lambda_2^I$ to empirically perform best overall.

Then, SOSP-I iteratively selects the structures to prune, based on the objective (2): Starting from an empty pruning mask $M = \{\}$, we iteratively add to $M$ the structure $s \notin M$ that minimizes the overall sensitivity $\lambda_2^I(M \cup \{s\})$. In practice, the algorithm pre-computes the matrix $Q \in \mathbb{R}^{S \times S}$,

$$Q_{s, s'} := \frac{1}{2} \left| \theta_s^T H(\theta) \theta_{s'} \right| + \left| \theta_s^T \frac{d\mathcal{L}(\theta)}{d\theta} \right| \cdot \delta_{s=s'}, \tag{3}$$

and selects at each iteration a structure $s \notin M$ to prune by

$$\operatorname*{arg\,min}_{s \notin M} \lambda_2^I(M \cup \{s\}) - \lambda_2^I(M) = \operatorname*{arg\,min}_{s \notin M} \left( Q_{s,s} + 2 \sum_{s' \in M} Q_{s,s'} \right), \tag{4}$$

terminating at the desired pruning ratio. To compute $Q$ efficiently, we show in App. B that the Hessian terms can be approximated as

$$\theta_s^T H(\theta) \theta_{s'} \approx \frac{1}{N'} \sum_{n=1}^{N'} \left( \phi(x_n) \theta_s \right)^T R_n \left( \phi(x_n) \theta_{s'} \right). \tag{5}$$

Herein, the NN gradient $\phi(x) := \nabla_\theta f_\theta(x) \in \mathbb{R}^{D \times P}$ forms the basis of the well-established Gauss-Newton approximation (see App. B for more details) used for the Hessian, and the matrices $R_n \in \mathbb{R}^{D \times D}$ are diagonal plus a rank-1 contribution (App. B). For further efficiency gains, the sum runs over a random subsample of size $N' < N$. In practice, one pre-computes all (sparse) products $\phi(x_n) \theta_s \in \mathbb{R}^D$ starting from the efficiently computable gradient $\phi(x_n)$, before aggregating a batch onto the terms $\theta_s^T H(\theta) \theta_{s'}$. Eq. (5) also has an interpretation as output correlations between certain network modifications, without derivatives (App. C).

## 2.2 SOSP-H: SALIENCY FROM HESSIAN-VECTOR PRODUCT

SOSP-H treats the second-order terms in (1) in a different way, motivated by the limit of large pruning ratios: At high pruning ratios, the sum $\sum_{s' \in M} \theta_{s'}$ in (1) can be approximated by $\sum_{s'=1}^{S} \theta_{s'} =: \theta_{struc}$ (this equals $\theta$ if every NN weight belongs to some structure $s$). The second-order term $\sum_{s,s' \in M} \theta_s^T H(\theta) \theta_{s'} \approx \left( \sum_{s \in M} \theta_s^T \right) \left( H(\theta) \theta_{struc} \right)$ thus becomes tractable since the Hessian-vector product $H(\theta)\theta_{struc}$ is efficiently computable by a variant of the backpropagation algorithm. To account for each structure $s$ and for the first- and second-order contributions separately, as above, we place absolute value signs in (1) so as to arrive at the final objective $\lambda_2^H(M) := \sum_{s \in M} \lambda_2^H(s)$ with

$$\lambda_2^H(s) := \left| \theta_s^T \frac{d\mathcal{L}(\theta)}{d\theta} \right| + \frac{1}{2} \left| \theta_s^T \left( H(\theta) \theta_{struc} \right) \right|. \tag{6}$$

To minimize $\lambda_2^H(M)$, SOSP-H starts from an empty pruning mask $M = \{\}$ and successively adds to $M$ a structure $s \notin M$ with smallest $\lambda_2^H(s)$, until the desired pruning ratio is reached.

Unlike the Gauss-Newton approximation in SOSP-I, SOSP-H uses the *exact* Hessian $H(\theta)$ in the Hessian-vector product. But SOSP-H can therefore not account for individual absolute $s$-$s'$-correlations, as some of those may cancel in the last term in Eq. (6), contrary to (2). Both SOSP-H and SOSP-I reduce to the same first-order pruning method when neglecting the second order, i.e. setting $H(\theta) := 0$; we compare to the resulting first-order method in Sec. 3.1 (see also App. A.1).

## 2.3 COMPUTATIONAL COMPLEXITY

We detail here the computational complexities of our methods (for the experimental evaluation see Sec. 3.2). The approximation of $Q$ in (3) requires complexity $O\left(N'D(F + P)\right) = O(N'DF)$ for computing all $\phi(x_n)\theta_s$, where $F \geq P$ denotes the cost of one forward pass through the network ($F \approx P$ for fully-connected NNs), plus $O(N'DS^2)$ for the sum in (5). This is tractable for modern NNs, while including the exact $H(\theta)$ would have complexity at least $O(N'DSF)$. Once $Q$ has been computed, the selection procedure based on (4) has overall complexity $O(S^3)$, which is feasible for most modern convolutional NNs (Sec. 3.2). The total complexity of the SOSP-I method is thus

$$O(N'DF) + O(N'DS^2) + O(S^3). \tag{7}$$

SOSP-H requires computational complexity $O(N'DF)$ both to compute all the first-order as well as all the second-order terms in Eq. (6). Together with the sorting of the saliency values $\lambda_2^H(s)$, SOSP-H has thus the same low overall complexity as a first-order method, namely

$$O(N'DF) + O(S \log(S)). \tag{8}$$

Due to its weak dependency on $S$, in practice, SOSP-H efficiently scales to large modern networks (see Sec. 3.2), and may even be used for *unstructured* second-order pruning, where $S = P$.

Both of our methods scale much better than naively including all off-diagonal Hessian terms, which is intractable for modern NNs due to its $O(N'DSF)$ scaling. Since SOSP-I builds on individual absolute sensitivities and on the established Gauss-Newton approximation, we use SOSP-I in the following in particular to validate our more efficient main method SOSP-H.

## 3 RESULTS

To evaluate our methods, we train and prune VGGs (Simonyan & Zisserman, 2014), ResNets (He et al., 2016), PlainNet (He et al., 2016), and DenseNets (Huang et al., 2017) on the Cifar10/100 (Krizhevsky et al., 2009) and ImageNet (Deng et al., 2009) datasets. Stochastic gradient descent with an initial learning rate of 0.1, a momentum of 0.9 and weight decay of $10^{-4}$ is used to train these networks. For ResNet-32/56 and VGG-Net on Cifar10/100, we use a batch size of 128, train for 200 epochs and reduce the learning rate by a factor of 10 after 120 and 160 epochs. For DenseNet-40 on Cifar10/100, we train for 300 epochs and reduce the learning rate after 150 and 225 epochs. To fine-tune the network after pruning, we exactly repeat this learning rate schedule. For ResNets on ImageNet, we use a batch size of 256, train for 128 epochs and use a cosine learning rate decay. For all networks, we prune feature maps (i.e. channels) from all layers except the last fully-connected layer; for ResNets, we also exclude the downsampling-path from pruning. We approximate the Hessians by a subsample of size $N' = 1000$ (see Sec. 2.1). We report the best or average final test accuracy over 3 trials if not noted otherwise.

Table 1: Comparison of SOSP to other global pruning methods for high pruning ratios. The comparison for moderate pruning ratios is deferred to the appendix (see App. A.3). We tuned our pruning ratios to similar values as reported by the referred methods. To ensure identical implementations of the network models in PyTorch, reference numbers are taken from Wang et al. (2019a) and Mingjie & Zhuang (2018). In accordance with all referred methods, we report the mean and standard deviation of the best accuracies observed during fine-tuning. For final accuracies after fine-tuning see App. A.13. * denotes the baseline model. Both SOSP methods perform either on par or outperform the competing global pruning methods.

| Dataset | Cifar10 | | | Cifar100 | | |
|---|---|---|---|---|---|---|
| Method | Test acc. (%) | Reduct. in weights (%) | Reduct. in MACs (%) | Test acc. (%) | Reduct. in weights (%) | Reduct. in MACs (%) |
| **VGG-Net*** | 94.18 | - | - | 73.45 | - | - |
| NN Slimming | 85.01 | 97.85 | 97.89 | 58.69 | 97.76 | 94.09 |
| NN Slim. $+L_1$ | 91.99 | 97.93 | 86.00 | 57.07 | 97.59 | 93.86 |
| C-OBD | $92.34 \pm 0.18$ | $97.68 \pm 0.02$ | $77.39 \pm 0.36$ | $58.07 \pm 0.60$ | $97.97 \pm 0.04$ | $77.55 \pm 0.25$ |
| EigenDamage | $92.29 \pm 0.21$ | $97.15 \pm 0.04$ | $86.51 \pm 0.26$ | $\mathbf{65.18} \pm 0.10$ | $97.31 \pm 0.01$ | $88.63 \pm 0.12$ |
| SOSP-I (ours) | $92.62 \pm 0.14$ | $97.79 \pm 0.02$ | $83.52 \pm 0.29$ | $64.20 \pm 0.23$ | $97.83 \pm 0.04$ | $87.02 \pm 0.20$ |
| SOSP-H (ours) | $\mathbf{92.71} \pm 0.19$ | $97.81 \pm 0.01$ | $86.32 \pm 0.29$ | $64.59 \pm 0.35$ | $97.81 \pm 0.01$ | $86.32 \pm 0.29$ |
| **ResNet-32*** | 95.30 | - | - | 76.8 | - | - |
| C-OBD | $91.75 \pm 0.42$ | $97.30 \pm 0.06$ | $93.50 \pm 0.37$ | $59.52 \pm 0.24$ | $97.74 \pm 0.08$ | $94.88 \pm 0.08$ |
| EigenDamage | $\mathbf{93.05} \pm 0.23$ | $96.05 \pm 0.03$ | $94.74 \pm 0.02$ | $65.72 \pm 0.04$ | $95.21 \pm 0.04$ | $94.62 \pm 0.06$ |
| SOSP-I (ours) | $92.43 \pm 0.09$ | $95.47 \pm 0.33$ | $94.07 \pm 0.66$ | $67.36 \pm 0.46$ | $92.69 \pm 0.07$ | $95.63 \pm 0.13$ |
| SOSP-H (ours) | $92.23 \pm 0.12$ | $95.26 \pm 0.10$ | $94.45 \pm 0.40$ | $\mathbf{68.42} \pm 0.21$ | $94.08 \pm 0.21$ | $95.06 \pm 0.14$ |
| **DenseNet-40*** | 94.58 | - | - | 74.11 | - | - |
| NN Slim. $+ L_1$ | 94.22 | 54.21 | - | $\mathbf{73.19}$ | 54.21 | - |
| SOSP-I (ours) | $94.21 \pm 0.04$ | $47.00 \pm 0.10$ | $36.35 \pm 0.12$ | $73.05 \pm 0.11$ | $45.22 \pm 0.10$ | $42.05 \pm 1.16$ |
| SOSP-H (ours) | $\mathbf{94.23} \pm 0.05$ | $49.39 \pm 0.65$ | $38.86 \pm 0.70$ | $73.05 \pm 0.24$ | $48.58 \pm 0.22$ | $42.05 \pm 0.35$ |

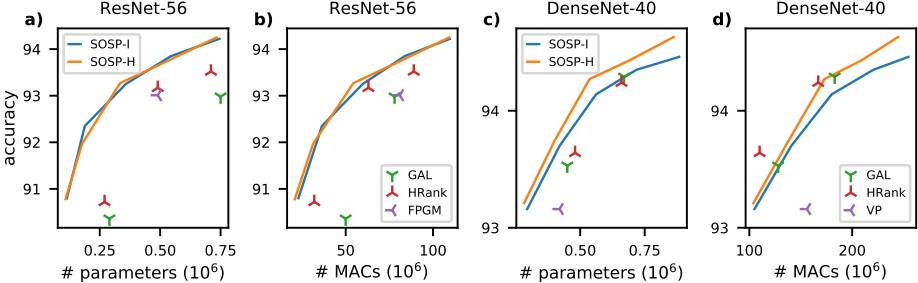

Figure 1: Comparison of SOSP to local, i.e. layer-wise, pruning methods on Cifar10 (see Suppl. Fig. 7 for a comparison to the CCP method). The best final test accuracy is plotted over the effective number of model parameters (a, c) and MACs (b, d). A tabular representation as well as statistics across trials are shown in App. A.14. SOSP-H especially outperforms all competing layer-wise pruning methods, especially over the number of effective parameters.

## 3.1 COMPARISON ON MEDIUM-SIZED DATASETS

In this section, we compare the performance of our SOSP methods and benchmark them against existing pruning algorithms on various medium-sized datasets and networks. First, we compare against other recent global pruning methods, then against pruning methods that learn layer-wise pruning ratios, and lastly against local structured pruning methods, i.e. with pre-specified layer-wise pruning ratios. For all comparisons we report the achieved test accuracy, the number of parameters of the pruned network, and the MACs (often referred to as FLOPs). To facilitate direct comparisons, we report test accuracies in the same way as the competing methods (e.g. best trial or average over trials), but additionally report mean and standard deviation of the test error for our models in App. A. Our count of the network parameters and MACs is based on the actual pruned network architecture (cf. App. D), even though our saliency measure associates with each structure only the weights *into* this structure (see Sec. 2).

Before comparing our SOSP methods to other recently published pruning algorithms, we perform ablation studies to motivate the use of our second-order objective which takes correlations into account. First, we compare the pruning accuracy of a first-order method (dropping all second-order terms in either Eq. (3) or (6)) to our second-order SOSP objectives. The results (see App. A.1) clearly

indicate, that second-order information improves the pruning performance. Second, we investigate whether the off-diagonal elements of the second-order terms improve the final accuracies. We perform this comparison using a variant of SOSP-I, where the off-diagonal terms $\sum_{s'} Q_{s,s'}$ in Eq. (4) are dropped. The results (see App. A.2) again suggest that adding these off-diagonal terms leads to a consistent improvement in final accuracy. In summary, we conclude that second-order pruning objectives that consider global correlations perform best.

We now first compare our SOSP methods to global pruning methods on VGG-Net, ResNet-32 and DenseNet-40. We use the same variants and implementations of these networks as used by Neural Network Slimming (NN Slimming; Liu et al., 2017) as well as EigenDamage and C-OBD (Wang et al., 2019a), e.g. capping the layer-wise ratio of removed structures at $95\%$ for VGGs to prevent layer collapse and increasing the width of ResNet-32 by a factor 4. C-OBD is a structured variant of the original OBD algorithm (Hassibi et al., 1993), which neglects all cross-structure correlations that, in contrast, SOSP takes into account. The results over three trials for high pruning ratios are shown in Tab. 1 and for moderate pruning ratios in App. A.3. To enable the comparison to NN Slimming on an already pretrained VGG, we included the results of NN Slimming applied to a baseline network obtained without modifications to its initial training, i.e. without $L_1$-regularization on the batch-normalization parameters. For moderate pruning ratios, all pruning schemes approximately retain the baseline performance for VGG-Net and ResNet-32 on Cifar10 and VGG-Net on Cifar100 (see Tab. 4). The only exception is the accuracy for C-OBD applied to VGG-Net on Cifar100, which drops by approximately $1\%$. For ResNet-32 on Cifar100 the accuracy after pruning is approximately $1\%$ lower than the baseline, for all pruning schemes. In the regime of larger pruning ratios of approximately $97\%$, SOSP and EigenDamage significantly outperform NN Slimming and C-OBD. SOSP performs on par with EigenDamage, except for ResNet-32 on Cifar100, where SOSP outperforms EigenDamage by almost $3\%$. For DenseNet-40, we achieve similar results compared to NN Slimming. However, note that NN Slimming depends on the above $L_1$-modification of network pretraining.

Next, we compare SOSP-H to AMC (He et al., 2018) which, instead of ranking the filters globally, learns layer-wise pruning ratios and then ranks filters within each layer. For a fair comparison, we extract their learned layer-wise pruning ratios for PlainNet-20 (He et al., 2016) and rank the filters in each layer according to their 2-norm (see also Han et al. (2015)). The results in App. A.4 show that SOSP performs better than AMC when compared over parameters, the main objective of SOSP, and on par with AMC when compared over MACs, the main objective of AMC.

Lastly, we compare our SOSP methods against four recently published local, i.e. layer-wise, pruning algorithms: FPGM He et al. (2019), GAL (Lin et al., 2019), CCP (Peng et al., 2019) (see Fig. 7), Variational Pruning (VP; Zhao et al., 2019) and HRank (Lin et al., 2020). For ResNet-56, our SOSP methods outperform all other methods across all pruning ratios (see Fig. 1a and b). For DenseNet-40, SOSP achieves better accuracies when compared over parameters (Fig. 1c) and is on par with the best other methods over MACs (Fig. 1d). The reason for this discrepancy is probably that the SOSP objective is agnostic to the number of MACs (image size) in each individual layer.

## 3.2 Scalability and application to large-scale datasets

Before going to large datasets, we compare the scalability of our methods SOSP-I and SOSP-H. As the preceding section shows, both methods perform basically on par with each other in terms of accuracy. This confirms that SOSP-H is not degraded by the approximations leading to the efficient Hessian-vector product, or is helped by use of the exact Hessian. In terms of efficiency, however, SOSP-H shows clear advantages compared to SOSP-I, for which the algorithm to select the structures to be pruned scales with $O(S^3)$ (see Sec. 2.3), potentially dominating the overall runtime for large-scale networks. Measurements of the actual runtimes show that already for medium-sized networks SOSP-H is more efficient than SOSP-I, see Fig. 2 and App. A.6. Since SOSP-I becomes impractical for large-scale networks like other second-order methods (Molchanov et al., 2019; Wang et al., 2019a), for the ImageNet dataset we will only evaluate SOSP-H and refer to it as SOSP.

On ImageNet, we compare our results to the literature for ResNet-18 and ResNet-50, see Tab. 2 or Suppl. Fig. 6. Because SOSP assesses the sensitivity of each structure independently of the contributed MACs, it has a bias towards pruning small-scale structures. This tendency is strongest for ResNet-50, due to its $1 \times 1$-convolutions. Since these $1 \times 1$-convolutions tend to contribute disproportionately little to the overall of MACs, we devised a scaled variant of SOSP, which divides

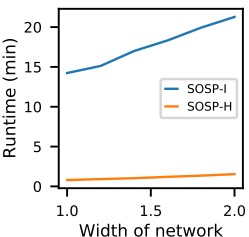

Figure 2: Runtime to calculate the pruning masks for ResNet-56 on Cifar10 over the width of the network for SOSP-I and SOSP-H. We vary the width of the network by increasing the width of each layer by a multiplicative factor.

Table 2: On ResNet-18/50 for ImageNet SOSP outperforms all competing methods. We compare the best final test accuracies and pruning ratios (PR) across 2 trials. A visualisation of the results can be found in App. A.7. For comparison to CCP, we also provide their alternative MAC count (for details, see App. D). * denotes SOSP with kernel scaling (see main text). "Gap" indicates the percentage gap to the respective baseline model.

| Model | Top-1% (Gap) | Parameters (PR) | MACs (PR) | Alt. MACs (PR) |
|---|---|---|---|---|
| **ResNet-18** | 69.76 (0.0) | 11.7M (0%) | 1.82B (0%) | 1.82B (0%) |
| SOSP (ours) | 69.63 (0.13) | 7.12M (39%) | 1.37B (24%) | 1.31B (28%) |
| FPGM | 68.41 (1.87) | 7.10M (39%) | 1.06B (41%) | - |
| SOSP (ours) | 68.78 (0.98) | 6.42M (45%) | 1.29B (29%) | 1.20B (34%) |
| **ResNet-50** | 76.15 (0.0) | 25.5M (0%) | 3.85B (0%) | 3.85B (0%) |
| SOSP (ours) | 76.56 (-0.41) | 19.9M (22%) | 3.06B (21%) | 2.72B (29%) |
| SOSP* (ours) | 76.60 (-0.45) | 17.9M (30%) | 2.79B (28%) | 2.47B (36%) |
| Taylor | 75.48 (0.67) | 17.9M (30%) | 2.66B (31%) | - |
| HRank | 74.98 (1.17) | 16.2M (36%) | 2.30B (44%) | - |
| FPGM | 75.59 (0.56) | 15.9M (37%) | 2.36B (42%) | - |
| SOSP (ours) | 75.85 (0.30) | 15.4M (40%) | 2.44B (27%) | 1.97B (49%) |
| Taylor | 74.50 (1.65) | 14.2M (44%) | 2.25B (37%) | - |
| HRank | 71.98 (4.17) | 13.8M (46%) | 1.55B (62%) | - |
| CCP | 75.21 (0.94) | - | - | 1.77B (54%) |
| SOSP* (ours) | 75.21 (0.94) | 13.0M (49%) | 2.13B (45%) | 1.68B (56%) |
| SOSP (ours) | 74.39 (1.76) | 11.8M (54%) | 1.89B (51%) | 1.38B (64%) |
| SOSP* (ours) | 73.38 (2.77) | 9.9M (61%) | 1.58B (59%) | 1.10B (72%) |

the saliency of every structure by its kernel size (e.g. 1, 3, or 7). Compared to the vanilla SOSP, the scaled variant of SOSP is able to remove larger percentages of MACs with similar drops in accuracy (see Tab. 2). We further show results on MobileNetV2 for Imagenet in the appendix (see App. A.5).

For both networks SOSP outperforms FPGM and HRank, especially when considering the main objective of SOSP, which is to reduce the number of parameters or structures. Since CCP uses a different way of calculating the MACs, which leads to consistently higher pruning ratios, we added an alternative MAC count to enable a fair comparison (for details, see App. D). Since HRank and FPGM do not mention their MAC counting convention, we assume they use the same convention as we do. Taking this into account, our scaled SOSP variant is able to prune more MACs than CCP, while having the same final accuracy. Further, we compare SOSP to Taylor (Molchanov et al., 2019), a global pruning method that uses only the first-order terms for its saliency ranking. SOSP clearly outperforms Taylor, even though both methods have the same scaling. This is also in line with the results of our ablation study (see Sec. 3.1 and esp. App. A.1), where we show that including second-order information leads to improved accuracies.

## 3.3 Identifying & Widening Architectural Bottlenecks

In the previous sections we evaluated the SOSP algorithm by removing those structures considered as least significant and then assessed the algorithm based on the performance of the pruned model. Here we instead follow an alternative path to validate the importance ranking introduced by SOSP, namely, we increase the size of these NN layers ranked as most significant. The idea for this comes from a common feature of the pruning masks found by SOSP for VGG and ResNet-56 (see Suppl. Fig. 8(d,g,e,h) and Fig. 3 (c,d,f,g)), namely that some layers are barely pruned while others are pruned by more than 80%. This could indicate towards architectural bottlenecks; widening these could further improve the performance of the model. We consider a layer an architectural bottleneck if it has a considerably lower pruning ratio compared to the other layers. Thus, widening these layers may improve the overall performance, and subsequent pruning could allow for even smaller models with higher accuracies compared to pruning the original model. To utilize this insight we devise a procedure that we call *expand-pruning*. The idea is to first calculate the pruning ratios of the trained network and then to identify architectural bottlenecks, i.e. the layers with the lowest pruning ratios. Next, we widen these specific layers by a factor of two, which has been proven to work well empirically. Finally, we randomly initialize, train, prune, and fine-tune the expanded network (for a schematic, see Fig. 3a). As a naive baseline that we call the *widen-pruning* procedure, we widen every layer in the network with a constant factor, instead of widening specific layers; this constant factor is chosen such that the overall number of parameters matches that of the expand-prune procedure (e.g., leading to the width multiplier 1.1 in Fig. 3).

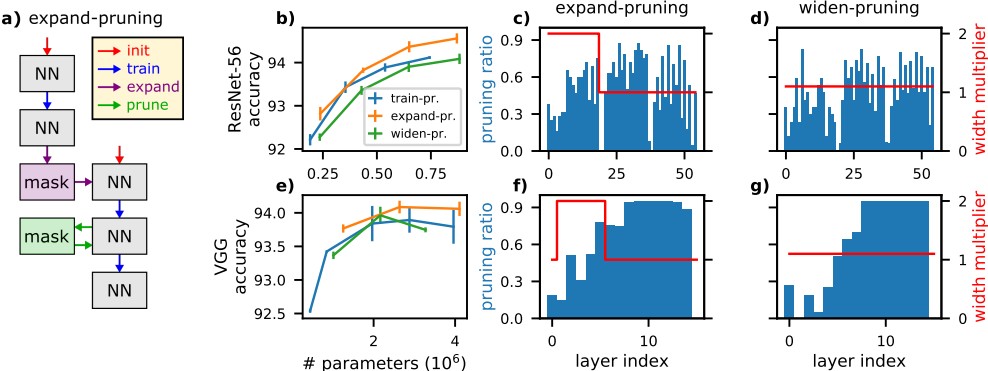

Figure 3: We remove architectural bottlenecks found by SOSP using the *expand-pruning* scheme (a) on Cifar10. The width of blocks and layers having low pruning ratios according to the standard pruning scheme ("train-pruning" in Suppl. Fig. 8d and g) are expanded by a width multiplier of 2 (c, f). As a baseline, we uniformly expand all layers in the network by a multiplier of 1.1 (d, g). The layer-wise pruning ratios of the enlarged network models are shown as bar plots in (c, d, f, g). The average test accuracy and standard deviation are shown over the number of model parameters (b, e).

We evaluate the expand-pruning procedure for ResNet-56 and VGG on Cifar10, for which we expand the least pruned of the three main building blocks and the five least pruned layers, respectively (cf. red *width multipliers* in Fig. 3(c,f) selected on the basis of the pruning masks shown in Suppl. Fig. 8(d,g)). Note that a more fine-grained widening of bottlenecks in ResNet-56, e.g. on layer level, is not possible without changing the overall ResNet architecture. In summary, a selective widening of bottlenecks results in smaller network models overall with higher accuracy than pruning the vanilla network or unselectively increasing the network size (compare expand-pruning and widen-pruning in Fig. 3(b,e)). While in principle any global pruning method could be used for the expand-pruning procedure, SOSP is especially suited since it does not require to modify the network architecture like EigenDamage. Further, SOSP can be applied directly at initialization (see App. A.10), unlike e.g. NN Slimming, allowing for a similar expand scheme directly at initialization (see App. A.11).

## 4 DISCUSSION

In this work we have demonstrated the effectiveness and scalability of our second-order structured pruning algorithms (SOSP). While both algorithms perform similarly well, SOSP-H is more easily scalable to large-scale networks and datasets. Therefore, we generally recommend to use SOSP-H over SOSP-I, especially for large-scale datasets and networks. The only regime where SOSP-I becomes a valuable alternative is for small pruning ratios and small- to medium-sized networks. Further, we showed that the pruning masks found by SOSP can be used to systematically detect and widen architectural bottlenecks, further improving the performance of pruned networks.

Compared to other global pruning methods, SOSP captures correlations between structures by a simple, effective and scalable algorithm that requires to modify neither the training nor the architecture of the network model to be pruned. SOSP achieves comparable or better accuracies on benchmark datasets, while having the same low complexity as first-order methods. The C-OBD algorithm (Wang et al., 2019a) is a structured generalization of the original unstructured OBD algorithm (LeCun et al., 1990). In contrast to OBD, C-OBD accounts for correlations within each structure, but does not capture correlations between different structures within and across layers. We show that considering these *global* correlations consistently improves the performance, especially for large pruning ratios (Tab. 1). This is in line with the results of our ablation study in App. A.2 (see also Sec. 3.1). The objective of EigenDamage (Wang et al., 2019a) to include second-order correlations is similar to ours, but the approaches are significantly different: EigenDamage uses the Fisher-approximation, which is similar to SOSP-I's Gauss-Newton approximation and exhibits a similar scaling behaviour, and then, in addition to further approximations, applies low-rank approximations that require the substitution of each layer by a bottleneck-block structure. Our SOSP method is simpler, easier to

implement and does not require to modify the network architecture, but nevertheless performs on par with EigenDamage. The approach of NN Slimming (Liu et al., 2017) is more heuristic than SOSP and is easy to implement. However, networks need to be pretrained with $L_1$-regularization on the batch-normalization parameters, otherwise the performance is severely harmed (Tab. 1 and Supp. Tab. 4). SOSP on the other hand does not require any modifications to the network training and thus can be applied to any network. A recent variant of NN Slimming was developed by Zhuang et al. (2020) who optimize their hyperparameters to reduce the number of MACs. Using the number MACs as an objective for SOSP is left for future studies. Most other existing global second-order structured pruning methods either approximate the Hessian as a diagonal matrix (Theis et al., 2018; Liu et al., 2021) or use an approximation similar to Gauss-Newton, which we used for SOSP-I (e.g., the Fisher information matrix in Molchanov et al., 2019). These approaches based on the Gauss-Newton approximation or its variants usually share the same disadvantageous scaling as that encountered by SOSP-I, and thus cannot scale to ImageNet as SOSP-H can.

In addition to the above comparison to other global pruning methods, we also compared our methods to simpler local pruning methods that keep the pruning ratios constant for each layer and, consequently, scale well to large-scale datasets. The pruning method closest to our SOSP-I method is the one by Peng et al. (2019). While both works consider second-order correlations between structures, theirs is based on a ranking scheme different from our absolute sensitivities in $\lambda_2^I$ and considers only intra-layer correlations. Furthermore, they employ an auxiliary classifier with a hyperparameter, yielding accuracy improvements that are difficult to disentangle from the effect of second-order pruning. Going beyond a constant pruning ratio for each layer, Su et al. (2020) discovered, for ResNets, that pruning at initialization seems to preferentially prune initial layers and thus proposed a pruning scheme based on a "keep-ratio" per layer which increases with the depth of the network. Our experiments confirm some of the findings of Su et al. (2020), but we also show that the specific network architectures found by pruning can drastically vary between different networks and especially between initialization and after training (histograms in Suppl. Fig. 8). While all local pruning methods specify pruning ratios for each layer, our method performs automatic selection across layers. Further, SOSP performs on par or better than AMC (He et al., 2018) which learns these layer-wise pruning ratios as a first step, instead of specifying them in advance like the above local methods. This is surprising since AMC uses many iterations to learn the layer-wise pruning ratios, while SOSP uses only a single step. Further, there exist even more elaborate schemes to find these subnetworks, in the spirit of AMC, such as Autocompress (Liu et al., 2020), EagleEye (Li et al., 2020), DMCP (Guo et al., 2020) and CafeNet (Su et al., 2021). It is difficult to enable a fair quantitative comparison between these iterative methods and our single-shot method SOSP.

SOSP's automatic selection of important structures allows us to identify and widen architectural bottlenecks. However, as the size of structures is usually identical within layers, our global pruning method has a bias towards pruning small structures, absent from local pruning methods. We proposed a simple solution to remove some of this bias by scaling each structure by the inverse of the kernel size. Alternatively, to better reflect the computational costs in real-world applications, each structure could also be normalized by the number of its required MACs or memory (van Amersfoort et al., 2020; Liu et al., 2021). Further, introducing additional hyperparameters for each layer optimized to reduce the number of MACs as done by Chin et al. (2020) could potentially lead to further improvements, but would also significantly increase the complexity due to the need of hyperparameter optimization. Liu et al. (2021) form groups of network structures that depend on each other, and compute the saliency for each group collectively, to better estimate the importance. While this proved to be beneficial, it is orthogonal to the investigation of finding good second-order approximations.

Recently, unstructured (Lee et al., 2018; Wang et al., 2019b; Tanaka et al., 2020) and structured (van Amersfoort et al., 2020; Hayou et al., 2021) pruning schemes were proposed that are applicable to networks at initialization. While these methods fail to achieve similar accuracies compared to pruning after training, we show in a small ablation study (see App. A.9) that our SOSP method at initialization can, after comparable training time, achieve accuracies similar to pruning after training.

In accordance with Elsken et al. (2019), our results suggest that pruning can be used to optimize the architectural hyperparameters of established networks (Liu et al., 2018) or super-graphs (Noy et al., 2020); see also App. A.12. We envision our second-order sensitivity analysis to be a valuable tool to identify and widen bottlenecks. For example, whenever a building block of the neural network cannot be compressed, this building block may be considered a bottleneck of the architecture and could be inflated to improve the overall trade-off between accuracy and computational cost.

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

Supplementary Material for

# SOSP: Efficiently Capturing Global Correlations by Second-Order Structured Pruning

## A  ADDITIONAL DATA AND EXPERIMENTS

### A.1  COMPARISON OF FIRST-ORDER STRUCTURED PRUNING WITH SOSP

See Figure 4 and Table 3 and their captions (cf. also Sec. 3.1 for further descriptions). The experiments are in line with our comparison to Taylor (Molchanov et al., 2019) on ImageNet, which is basically a first-order method (see Sec. 3.2).

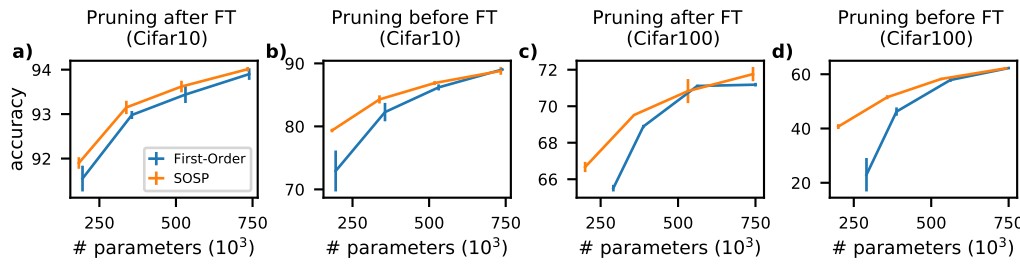

Figure 4:  Comparison of the accuracies achieved by SOSP and a first-order method, where all second-order terms are set to zero (see end of Sec. 2.2), for ResNet-56 before and after the fine-tuning step on Cifar10 and Cifar100. All shown experiments prune a full ResNet-56 model with $855 \times 10^3$ parameters; the parameter count of the pruned network is shown on the horizontal axis. The results clearly show that adding second-order information increases the performance, before as well as after fine-tuning (FT). The advantage of using second-order information increases as the size of the pruned network decreases.

Table 3:  Comparison of SOSP to first-order global pruning for ResNet-56 on Cifar10 and Cifar100.

| Dataset | Cifar10 | | | | Cifar100 | | | |
|---|---|---|---|---|---|---|---|---|
| Method (Pruning Ratio) | Test acc. (%) | Test acc. before FT (%) | Numb. of Params. ($10^3$) | Numb. of MACs ($10^7$) | Test acc. (%) | Test acc. before FT (%) | Numb. of Params. ($10^3$) | Numb. of MACs ($10^7$) (%) |
| First-Order (0.1) | $93.90 \pm 0.13$ | $89.03 \pm 0.09$ | 739 | 108 | $71.18 \pm 0.1$ | $62.27 \pm 0.40$ | 750 | 105 |
| SOSP (0.1) | $94.01 \pm 0.05$ | $88.81 \pm 0.62$ | 734 | 109 | $71.76 \pm 0.39$ | $62.19 \pm 0.24$ | 742 | 107 |
| First-Order (0.3) | $93.44 \pm 0.19$ | $86.19 \pm 0.48$ | 531 | 76 | $71.11 \pm 0.07$ | $57.88 \pm 0.70$ | 562 | 71 |
| SOSP (0.3) | $93.62 \pm 0.13$ | $86.88 \pm 0.31$ | 518 | 79 | $70.88 \pm 0.66$ | $58.25 \pm 0.46$ | 532 | 77 |
| First-Order (0.5) | $92.98 \pm 0.09$ | $82.27 \pm 1.46$ | 356 | 50 | $68.89 \pm 0.09$ | $46.24 \pm 1.54$ | 388 | 45 |
| SOSP (0.5) | $93.15 \pm 0.15$ | $84.28 \pm 0.65$ | 338 | 55 | $69.52 \pm 0.03$ | $51.53 \pm 0.75$ | 358 | 51 |
| First-Order (0.7) | $91.55 \pm 0.29$ | $72.92 \pm 3.25$ | 195 | 28 | $65.51 \pm 0.18$ | $22.96 \pm 6.16$ | 291 | 23 |
| SOSP (0.7) | $91.90 \pm 0.13$ | $79.35 \pm 0.28$ | 183 | 31 | $66.67 \pm 0.28$ | $40.71 \pm 0.91$ | 200 | 29 |

### A.2  COMPARISON OF ACCURACIES ACHIEVED BY SOSP-I WITH AND WITHOUT CROSS-STRUCTURE CORRELATIONS

See Figure 5 and its caption (cf. also Sec. 3.1 for further descriptions).

### A.3  RESULTS FOR MEDIUM PRUNING RATES FOR COMPARING GLOBAL PRUNING METHODS

See Table 4 and its caption. (For high pruning rates, see Table 1 in Sec. 3.1.)

### A.4  COMPARISON TO PRUNING-METHODS THAT FIND LAYER-WISE PRUNING RATES

See Table 5 and its caption (cf. also Sec. 3.1 for further descriptions).

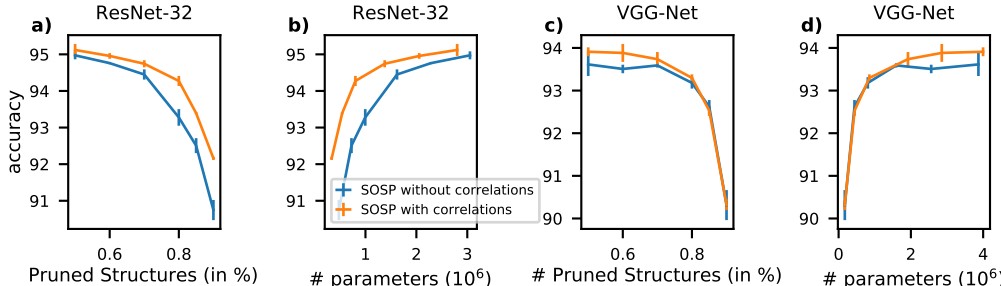

Figure 5: Comparison of the accuracies achieved by vanilla SOSP-I and a variation of SOSP-I, where all off-diagonal terms of the Hessian are set to zero, for ResNet-56 and VGG on Cifar10. The results on both networks suggest that cross-structure correlations can significantly improve pruning performance. The gap in performance is even larger before the fine-tuning step (not shown).

Table 4: Comparison of SOSP to other global pruning methods for moderate pruning rates. The setting is exactly the same as in Tab. 1. For final accuracies after fine-tuning see App. A.13. * denotes the baseline model.

| Dataset | Cifar10 | | | Cifar100 | | |
|---|---|---|---|---|---|---|
| Method | Test acc. (%) | Reduct. in weights (%) | Reduct. in MACs (%) | Test acc. (%) | Reduct. in weights (%) | Reduct. in MACs (%) |
| **VGG-Net*** | 94.18 | - | - | 73.45 | - | - |
| NN Slimming | 92.84 | 80.07 | 42.65 | 71.89 | 74.60 | 38.33 |
| NN Slim. $+L_1$ | 93.79 | 83.45 | 49.23 | 72.78 | 76.53 | 39.92 |
| C-OBD | **94.04** $\pm$ 0.12 | 82.01 $\pm$ 0.44 | 38.18 $\pm$ 0.45 | 72.23 $\pm$ 0.15 | 77.03 $\pm$ 0.05 | 33.70 $\pm$ 0.04 |
| EigenDamage | 93.98 $\pm$ 0.06 | 78.18 $\pm$ 0.12 | 37.13 $\pm$ 0.41 | 72.90 $\pm$ 0.06 | 76.64 $\pm$ 0.12 | 37.40 $\pm$ 0.11 |
| SOSP-I (ours) | 93.99 $\pm$ 0.17 | 85.75 $\pm$ 0.74 | 45.96 $\pm$ 4.29 | **73.17** $\pm$ 0.11 | 82.68 $\pm$ 0.04 | 44.87 $\pm$ 0.61 |
| SOSP-H (ours) | 93.73 $\pm$ 0.16 | 87.29 $\pm$ 0.21 | 57.74 $\pm$ 2.57 | 73.11 $\pm$ 0.19 | 79.20 $\pm$ 0.35 | 51.61 $\pm$ 0.98 |
| **ResNet-32*** | 95.30 | - | - | 76.8 | - | - |
| C-OBD | 95.11 $\pm$ 0.10 | 70.36 $\pm$ 0.39 | 66.18 $\pm$ 0.46 | **75.70** $\pm$ 0.31 | 66.68 $\pm$ 0.25 | 67.53 $\pm$ 0.25 |
| EigenDamage | 95.17 $\pm$ 0.12 | 71.99 $\pm$ 0.13 | 70.25 $\pm$ 0.24 | 75.51 $\pm$ 0.11 | 69.80 $\pm$ 0.11 | 71.62 $\pm$ 0.21 |
| SOSP-I (ours) | 95.06 $\pm$ 0.07 | 72.33 $\pm$ 0.50 | 67.36 $\pm$ 0.80 | 75.33 $\pm$ 0.11 | 63.83 $\pm$ 0.17 | 74.28 $\pm$ 0.08 |
| SOSP-H (ours) | **95.22** $\pm$ 0.12 | 72.85 $\pm$ 0.40 | 67.85 $\pm$ 0.37 | 75.52 $\pm$ 0.20 | 69.31 $\pm$ 0.36 | 71.60 $\pm$ 0.38 |
| **DenseNet-40*** | 94.58 | - | - | 74.11 | - | - |
| NN Slim. $+ L_1$ | 94.32 | 35.52 | - | **73.76** | 35.45 | - |
| SOSP-I (ours) | **94.42** $\pm$ 0.03 | 32.21 $\pm$ 0.16 | 22.03 $\pm$ 0.13 | 73.46 $\pm$ 0.05 | 31.38 $\pm$ 0.09 | 29.98 $\pm$ 0.55 |
| SOSP-H (ours) | 94.41 $\pm$ 0.12 | 34.78 $\pm$ 0.67 | 26.14 $\pm$ 0.13 | 73.60 $\pm$ 0.17 | 34.15 $\pm$ 0.13 | 28.23 $\pm$ 0.09 |

## A.5 COMPARISON OF SOSP ON MOBILENETV2 FOR IMAGENET

See Table 6 for a comparison of SOSP for MobileNetV2 for Imagenet.

## A.6 RUNTIME COMPARISONS OF SOSP-H AND SOSP-I

The results of Table 7 complement the results of Fig. 2 and show that while SOSP-I is still practical for small-scale networks and datasets for large scale networks and datasets it becomes impractical. SOSP-H on the other side says practical across all network dataset sizes.

## A.7 PLOTS FOR COMPARISON OF SOSP ON RESNET-18/50 FOR LARGE-SCALE DATASETS

See Figure 6 and its caption (cf. also Sec. 3.2 for further descriptions).

Table 5: We compare the results of SOSP and AMC for PlainNet-20 on Cifar10. We show the mean and standard deviation of the accuracy over three independent runs.

| Method | Acc. | Params ($10^3$) | MACs ($10^6$) |
|--------|------|-----------------|----------------|
| AMC | 90.23±0.10 | 132 | 20.9 |
| SOSP (0.3) | 90.93±0.17 | 124 | 26.7 |
| SOSP (0.4) | 90.14±0.15 | 90 | 22.7 |

Table 6: We compare the results of SOSP with other state-of-the-art methods on MobileNetV2 for ImageNet. "Gap" indicates the percentage gap to the respective baseline model and PR stands for pruning ratio.

| Method | Top-1% (Gap) | Parameters (PR) | MACs (PR) |
|--------|--------------|-----------------|-----------|
| MobileNetV2 | 71.88(0.00) | 3.5M(0%) | 0.30B(0%) |
| AMC | 70.80(1.08) | – | 0.21B(30%) |
| SOSP (0.2) | **71.04(0.84)** | 2.5M(28%) | 0.21B(29%) |

## A.8 COMPARISON OF SOSP ON RESNET-56 AGAINST CCP

See Figure 7 and its caption (cf. also Sec. 3 for further comparisons on ResNet-56).

## A.9 PRUNING AT INITIALIZATION

Traditionally, pruning methods are applied to pretrained networks, as also done in the main part of our work, but recently there has been growing attention on pruning at initialization following the works of Lee et al. (2018) and Frankle & Carbin (2018). Since SOSP employs the absolute value of the sensitivities, it can also be applied to a randomly initialized network without any modifications. Thus, SOSP can also be seen as an efficient second-order generalization of SNIP (Lee et al., 2018; van Amersfoort et al., 2020). While EigenDamage can in principle be modified and applied to a randomly initialized network as well, NN Slimming can not be applied at initialization.

Usually pruning at initialization leads to worse accuracies than pruning after training (Liu et al., 2018). However, pruning an already trained network is often followed by fine-tuning, effectively training the network twice (Fig. 8a). For comparability with pruning at initialization, we unify the overall training schedule between these two settings and consequently apply two training cycles after pruning the randomly initialized network (Fig. 8b; for further discussion, see App. A.10).

We observe that applying SOSP at initialization performs almost equally well than applying SOSP after training (see ResNet-56 and VGG in Fig. 8c and f, respectively). In conclusion, applying SOSP at initialization can significantly reduce the time and resources required for network training with no or only minor degradation in accuracy.

## A.10 PRUNING AT INITIALIZATION VS PRUNING A PRETRAINED NEURAL NETWORK

In this section we investigate further why pruning a randomly initialized network tends to achieve lower accuracies compared to pruning a pretrained network (Lee et al., 2018; van Amersfoort et al., 2020). We compare the final accuracies of pruning and fine-tuning a randomly initialized and pretrained network (see Fig. 9). While pruning a pretrained network leads to considerably higher accuracies compared to pruning a randomly initialized network, the random baseline curves show the same difference. Thus, to be able to compare pruning before and after training one needs to device settings that enable a fair comparison, ensuring similar accuracies for random pruning accross both settings.

Table 7: Comparison of time needed to calculate the importance vector for SOSP-I and SOSP-H for several networks on Cifar10 and ImageNet.

| Time needed for pruning step (without training) | ResNet-56 | ResNet-32 | DenseNet-40 | VGG | ResNet50 (Imagenet) |
|---|---|---|---|---|---|
| SOSP-H | $50 \pm 2$ s | $123\pm4$s | $67\pm2$s | $110\pm3$s | $40\pm2$ min |
| SOSP-I | $926\pm26$s | $1355\pm38$s | $635\pm18$s | $1647\pm44$s | $> 48$h |

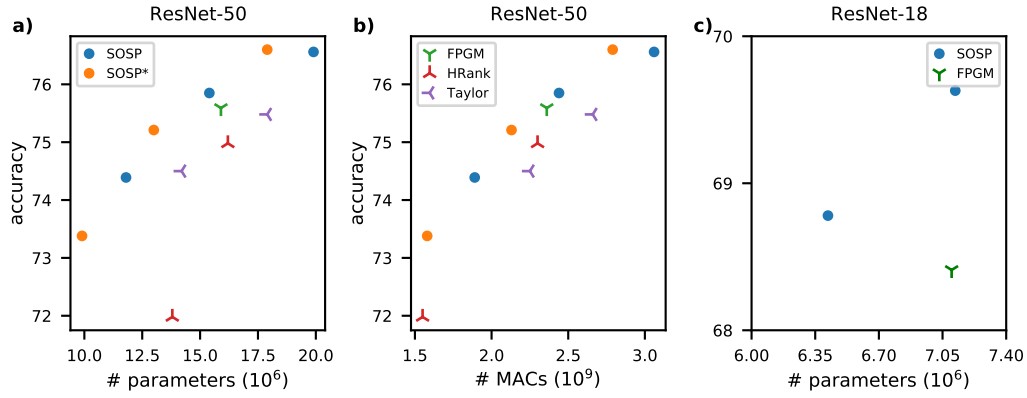

Figure 6: Comparison of the accuracies achieved by SOSP and its competing methods on ResNet-18/50 on ImageNet. The plots visualize the data of Tab. 2.

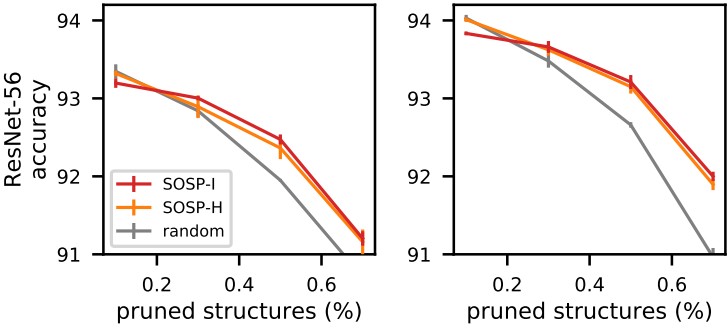

Figure 9: Comparison of pruning and fine-tuning a randomly initialized (left) and a pretrained (right) network for ResNet-56 on Cifar10. Random pruning is a baseline which selects uniformly at random structures $s$ and adds them to the mask $M$ until the predefined pruning ratio is reached.

## A.11 EXPAND-INIT DATA

This section provides the experimental results of the corresponding expand-procedure at initialization (see Sec. 3.3). The results of Fig. 8 indicate that architectural bottlenecks exist not only for pretrained networks but also for randomly initialized networks. Therefore, we device also an expand scheme before training. In this scheme the mask for the expand-procedure is not calculated for a pretrained network but for a randomly initialized network. Following the expand scheme the network is pruned and then only fine-tuned once. The results are displayed in Fig. 10.

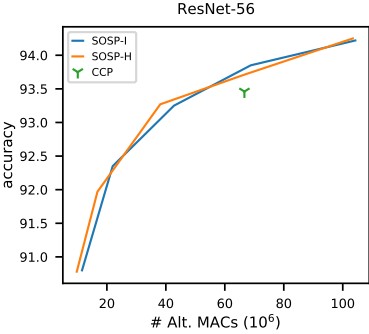

Figure 7: Comparison of the accuracies for Cifar10 achieved by SOSP-H and SOSP-I vs. CCP (Peng et al., 2019) over the alternative MAC-count used by (Peng et al., 2019) (see also App. D).

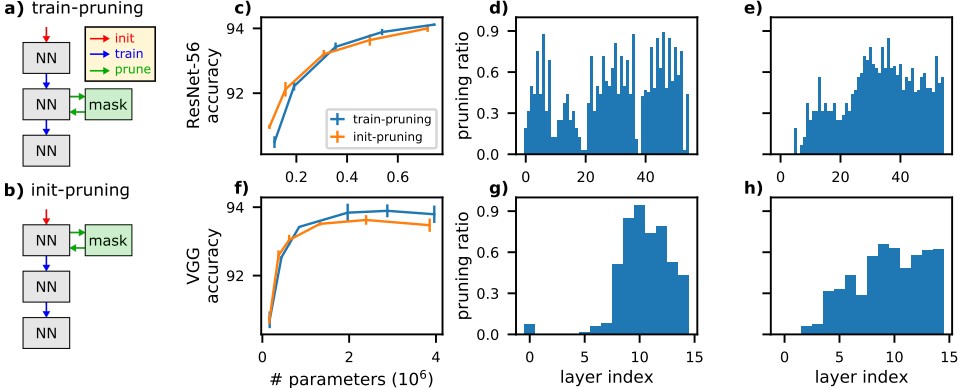

Figure 8: Comparison between pruning after training and at initialization on Cifar10. Both pruning schemes, train-pruning (a) and init-pruning (b), train the network for the same overall number of epochs, but generate and apply the pruning masks at different point in times. The average and standard deviation of the test accuracy across 3 trials is plotted against the number of model parameters for ResNet-56 (top row; c) and VGG (bottom row; f). For a single trial, in which overall 50% of the structures are pruned, we visualize the pruning masks of train-pruning and init-pruning by showing the layer-wise pruning ratios in (d, g) and (e, h), respectively.

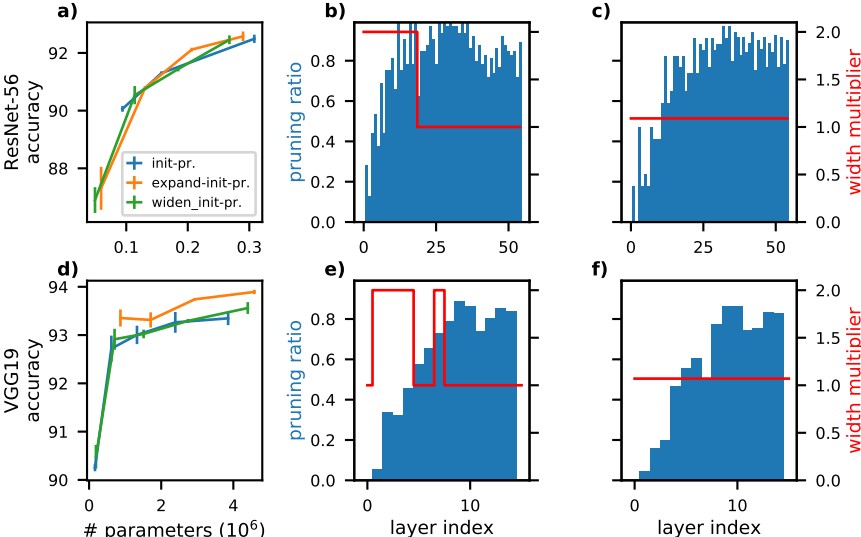

Figure 10: We widen architectural bottlenecks found by SOSP starting with a randomly initalized network. The width of blocks and layers with low pruning ratios in the train-pruning scheme (Fig. 8e and h) are expanded by a width multiplier of 2 (b, e). As a baseline, we again uniformly expand all layers in the network by a factor 1.1 (c, f). The layer-wise pruning ratios of the enlarged network models are shown as bar plots in (b, c, e, f). The average and standard deviation of the test accuracy

## A.12 Shuffle Experiments

See Table 8 and its caption.

Table 8: We compare the results of SOSP with Shuffle SOSP, where the pruning masks of the structures within each layer, and therefore also the weights, are shuffled before the fine-tuning step. Therefore, Shuffle SOSP retains the same architecture (the same number of structures are pruned per layer) as SOSP, but prunes different structures within each layer. The results indicate that the dominant factor is the architecture and not he specific values of the weights.

| Structured Pruning Ratio | Acc. SOSP | Acc.SOSP-Shuffle |
|---|---|---|
| 0.3 | $93.62 \pm 0.13$ | $92.59 \pm 0.12$ |
| 0.5 | $93.15 \pm 0.15$ | $92.94 \pm 0.09$ |

## A.13 Mean and Standard Deviation of Final Accuracies for the Global Pruning Comparison

See Table 9 and its caption.

Table 9: Mean and standard deviation of the final accuracies after the full fine-tuning step of both SOSP methods on Cifar10 and Cifar100 for VGG-Net, ResNet-32 and DenseNet-40.

| Dataset | CIFAR10 | | | | | | CIFAR100 | | | | | |
|---|---|---|---|---|---|---|---|---|---|---|---|---|
| Pruning Ratio | moderate | | | high | | | moderate | | | high | | |
| Method | Test acc (%) | Reduction in weights (%) | Reduction in MACs (%) | Test acc (%) | Reduction in weights (%) | Reduction in MACs (%) | Test acc (%) | Reduction in weights (%) | Reduction in MACs (%) | Test acc (%) | Reduction in weights (%) | Reduction in MACs (%) |
| **VGG-Net(Baseline)** | 94.18 | - | - | - | - | - | 73.45 | - | - | - | - | - |
| SOSP-I | $93.88 \pm 0.21$ | $85.75 \pm 0.74$ | $45.96 \pm 4.29$ | $92.53 \pm 0.13$ | $97.79 \pm 0.02$ | $83.52 \pm 0.29$ | $72.93 \pm 0.28$ | $82.68 \pm 0.04$ | $44.87 \pm 0.61$ | $63.87 \pm 0.06$ | $97.83 \pm 0.04$ | $87.02 \pm 0.20$ |
| SOSP-H | $93.65 \pm 0.16$ | $87.29 \pm 0.21$ | $57.74 \pm 2.57$ | $92.59 \pm 0.19$ | $97.81 \pm 0.01$ | $86.32 \pm 0.29$ | $72.94 \pm 0.28$ | $79.20 \pm 0.35$ | $51.61 \pm 0.98$ | $64.24 \pm 0.55$ | $97.81 \pm 0.01$ | $86.32 \pm 0.29$ |
| **ResNet-32(Baseline)** | 95.30 | - | - | - | - | - | 76.8 | - | - | - | - | - |
| SOSP-I | $94.96 \pm 0.08$ | $72.33 \pm 0.50$ | $67.36 \pm 0.80$ | $92.19 \pm 0.07$ | $95.47 \pm 0.33$ | $94.07 \pm 0.66$ | $75.10 \pm 0.10$ | $63.83 \pm 0.17$ | $74.28 \pm 0.08$ | $65.97 \pm 0.52$ | $92.69 \pm 0.07$ | $95.63 \pm 0.13$ |
| SOSP-H | $95.17 \pm 0.12$ | $72.85 \pm 0.40$ | $67.85 \pm 0.37$ | $91.97 \pm 0.04$ | $95.26 \pm 0.10$ | $94.45 \pm 0.40$ | $75.39 \pm 0.27$ | $69.31 \pm 0.36$ | $71.60 \pm 0.38$ | $67.37 \pm 0.26$ | $94.08 \pm 0.21$ | $95.06 \pm 0.14$ |
| Pruning Ratio | moderate | | | high | | | moderate | | | high | | |
| **DenseNet-40(Baseline)** | 94.58 | - | - | - | - | - | 74.11 | - | - | - | - | - |
| SOSP-I | $94.29 \pm 0.04$ | $32.21 \pm 0.16$ | $22.03 \pm 0.13$ | $94.07 \pm 0.07$ | $47.00 \pm 0.10$ | $36.35 \pm 0.12$ | $72.47 \pm 0.47$ | $31.38 \pm 0.09$ | $29.98 \pm 0.55$ | $72.10 \pm 0.10$ | $45.22 \pm 0.10$ | $42.05 \pm 1.16$ |
| SOSP-H | $94.28 \pm 0.11$ | $34.78 \pm 0.67$ | $26.14 \pm 0.13$ | $94.15 \pm 0.08$ | $49.39 \pm 0.65$ | $38.86 \pm 0.70$ | $72.88 \pm 0.43$ | $34.15 \pm 0.13$ | $28.23 \pm 0.09$ | $72.09 \pm 0.44$ | $48.58 \pm 0.22$ | $42.05 \pm 0.35$ |

## A.14 Tabular Data and Mean and Standard Deviation for Layer-Wise Pruning Comparison

This section provides additional data complementing the results of Fig. 1. The numerical data of Fig. 1 is shown in Tab. 10 and the mean and standard deviation of the final accuracies for both SOSP algorithms is shown in Tab. 11 and Fig. 11.

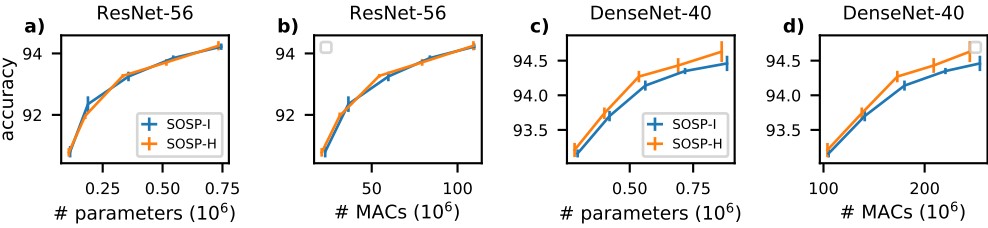

Figure 11: Mean and standard deviation plots of the accuracies after fine-tuning of SOSP for ResNet-56 and DenseNet-40 on Cifar10.

Table 10: Pruning results of ResNet-56 and DenseNet-40 on Cifar10. *Gap* denotes the difference between the accuracy of the pruned model and the baseline accuracy. *PR* denotes the pruning ratio, i.e. the percentage drop in MACs or parameters.

| Model | Top-1(Gap)% | Parameters(PR) | MACs(PR) |
|---|---|---|---|
| **ResNet-56** | 93.88(0.0) | 0.85M(0%) | 125M(0%) |
| GAL Lin et al. (2019) | 92.98(0.28) | 0.75M(12%) | 78M(38%) |
| SOSP-I (ours) | 94.22(-0.44) | 0.74M (13%) | 110M (13%) |
| SOSP-H (ours) | 94.25(-0.47) | 0.73M (15%) | 109M (14%) |
| HRank Lin et al. (2020) | 93.52(-0.26) | 0.71M(17%) | 89M(29%) |
| SOSP-I (ours) | 93.85(0.03) | 0.54M (36%) | 84M (33%) |
| SOSP-H (ours) | 93.71(0.17) | 0.52M (40%) | 79M (37%) |
| FPGM He et al. (2019) | 93.01(0.58) | 0.49M(42%) | 81M(36%) |
| HRank Lin et al. (2020) | 93.17(0.09) | 0.49M(42%) | 63M(50%) |
| SOSP-I (ours) | 93.25(0.53) | 0.36M (58%) | 60M (53%) |
| SOSP-H (ours) | 93.27(0.51) | 0.33M (61%) | 54M (57%) |
| GALLin et al. (2019) | 90.36(2.10) | 0.29M(66%) | 50M(60%) |
| HRank Lin et al. (2020) | 90.72(2.54) | 0.27M(68%) | 32M(74%) |
| SOSP-I (ours) | 92.35(1.53) | 0.19M (78%) | 36M (71%) |
| SOSP-H (ours) | 91.97(1.91) | 0.18M(79%) | 31M (75%) |
| SOSP-I (ours) | 90.80(3.08) | 0.11M (87%) | 22M (82%) |
| SOSP-H (ours) | 90.78(3.10) | 0.11M(87%) | 21M (83%) |
| **DenseNet-40** | 94.58(0.0) | 1.04M(0%) | 283M(0%) |
| SOSP-I (ours) | 94.46(0.12) | 0.88M(17%) | 255M(10%) |
| SOSP-H (ours) | 94.63(-0.05) | 0.86M(19%) | 245M(14%) |
| GAL Lin et al. (2019) | 94.29(0.52) | 0.67M(36%) | 183M(35%) |
| HRank Lin et al. (2020) | 94.24(0.57) | 0.66M(37%) | 167M(41%) |
| SOSP-I (ours) | 94.35(0.22) | 0.72M(32%) | 221M(22%) |
| SOSP-H (ours) | 94.43(0.15) | 0.69M(35%) | 209M(26%) |
| SOSP-I (ours) | 94.14(0.44) | 0.56M(47%) | 180M(36%) |
| SOSP-H (ours) | 94.27(0.31) | 0.54M(49%) | 172M(39%) |
| HRank Lin et al. (2020) | 93.68(1.13) | 0.48M(54%) | 110M(61%) |
| GAL Lin et al. (2019) | 93.53(1.28) | 0.45M(57%) | 128M(55%) |
| Zhao et al. (2019) | 93.16(0.95) | 0.42M(60%) | 156M(45%) |
| SOSP-I (ours) | 93.70(0.88) | 0.42M(60%) | 141M(50%) |
| SOSP-H (ours) | 94.74(0.84) | 0.40M(62%) | 138M(51%) |

Table 11: Mean and standard deviations of the accuracies after fine-tuning of SOSP for ResNet-56 and DenseNet-40.

| Model | Top-1% | Pruned Parameters (in %) | Pruned MACs (in %) |
|---|---|---|---|
| **ResNet-56** | 93.39 | 0 | 0 |
| SOSP-I | $94.22 \pm 0.10$ | $12.95 \pm 0.15$ | $12.78 \pm 0.09$ |
| SOSP-H | $94.25 \pm 0.14$ | $14.90 \pm 0.12$ | $13.11 \pm 0.36$ |
| SOSP-I | $93.85 \pm 0.09$ | $36.36 \pm 0.11$ | $33.31 \pm 0.12$ |
| SOSP-H | $93.71 \pm 0.11$ | $39.70 \pm 0.39$ | $36.86 \pm 0.70$ |
| SOSP-I | $93.25 \pm 0.16$ | $58.19 \pm 0.28$ | $52.53 \pm 0.21$ |
| SOSP-H | $93.27 \pm 0.06$ | $60.95 \pm 0.29$ | $56.74 \pm 0.31$ |
| SOSP-I | $92.35 \pm 0.25$ | $77.98 \pm 0.46$ | $71.04 \pm 0.21$ |
| SOSP-H | $91.97 \pm 0.11$ | $79.29 \pm 0.17$ | $75.07 \pm 0.62$ |
| SOSP-I | $90.8 \pm 0.18$ | $86.68 \pm 0.14$ | $81.79 \pm 0.23$ |
| SOSP-H | $90.78 \pm 0.14$ | $87.31 \pm 0.29$ | $83.42 \pm 0.15$ |
| **DenseNet-40** | 94.16 | 0 | 0 |
| SOSP-I | $94.46 \pm 0.11$ | $16.59 \pm 0.22$ | $10.00 \pm 0.72$ |
| SOSP-H | $94.63 \pm 0.15$ | $18.57 \pm 0.63$ | $13.52 \pm 2.13$ |
| SOSP-I | $94.35 \pm 0.04$ | $32.21 \pm 0.16$ | $22.03 \pm 0.13$ |
| SOSP-H | $94.43 \pm 0.11$ | $34.78 \pm 0.67$ | $26.14 \pm 1.16$ |
| SOSP-I | $94.14 \pm 0.07$ | $47.00 \pm 0.10$ | $36.35 \pm 0.12$ |
| SOSP-H | $94.27 \pm 0.08$ | $49.39 \pm 0.65$ | $38.86 \pm 0.70$ |
| SOSP-I | $94.70 \pm 0.08$ | $60.32 \pm 0.31$ | $50.24 \pm 0.80$ |
| SOSP-H | $94.74 \pm 0.08$ | $62.24 \pm 0.74$ | $51.28 \pm 1.07$ |

# B    APPROXIMATIVE SECOND-ORDER DERIVATIVES

In order to approximate the Hessian terms $\theta_s^T H(\theta)\theta_{s'}$ in the matrix $Q$ efficiently (see Eq. (2) and (3)), we omit from $H(\theta) = \frac{1}{N} \sum_n \nabla_\theta^2 \ell(f_\theta(x_n), y_n)$ those terms that involve the expensive second-order derivatives $\nabla_\theta^2 f_\theta(x_n)$ of the NN outputs, while including second-order couplings due to $\ell$. This is equivalent to approximating $H(\theta) \approx H(f_\theta^{lin}) := \frac{1}{N} \sum_n \nabla_\theta^2 \ell(f_\theta^{lin}(x_n), y_n)$ for the linearized

$f_{\theta'}(x) \approx f_{\theta'}^{lin}(x) := f_\theta(x) + \phi(x) \cdot (\theta' - \theta)$ with $\phi(x) := \nabla_\theta f_\theta(x) \in \mathbb{R}^{D \times P}$, which is well motivated by the NTK limit (Jacot et al., 2018) for large NNs at both initialization and after training. The terms in the sum become then

$$\nabla_\theta^2 \ell \left( f_\theta^{lin}(x_n), y_n \right) = \phi(x_n)^T R_n \phi(x_n), \tag{9}$$

where $R_n \in \mathbb{R}^{D \times D}$ is diagonal for squared loss, and has an additional rank-1 contribution for cross-entropy (see App. B.1 and App. B.2, respectively). Similar Hessian approximations were employed before in NNs (Hassibi et al., 1993; Wang et al., 2019a; Peng et al., 2019; Liu et al., 2021; Theis et al., 2018) and also in the Gauss-Newton optimization method (Fletcher, 2013).

Below we derive our approximation of the second-order loss derivatives Eq. (9) that lead to the desired Eq. (5), and especially the particular matrix structures of $R_n$ in these expressions (diagonal plus rank-1), which allow for a more efficient matrix multiplication (App. B.4).

Our approximation is based on omitting from the exact loss derivative those terms that involve the (expensive) second-order derivatives $\nabla_\theta^2 f_\theta(x_n)$ of the NN outputs $f_\theta(x_n) \in \mathbb{R}^D$. We however still include second-order couplings due to the loss function $\ell$. Our approximation is thus to approximate the NN output

$$f_{\theta'}(x) \approx f_{\theta'}^{lin}(x) := f_\theta(x) + \phi(x) \cdot (\theta' - \theta) \tag{10}$$

to first order, where $\phi(x) := \nabla_\theta f_\theta(x) \in \mathbb{R}^{D \times P}$ is the first-order derivative of the NN output, and then to approximate the second-order derivatives of the NN loss as follows:

$$\nabla_\theta^2 \ell \left( f_\theta(x_n), y_n \right) \approx \nabla_\theta^2 \ell \left( f_\theta^{lin}(x_n), y_n \right). \tag{11}$$

We now compute this approximation $\nabla_\theta^2 \ell \left( f_\theta^{lin}(x_n), y_n \right)$ for both the *squared loss* $\ell(f, y) := \frac{1}{2} \|f - y\|^2$ as well as for the *cross-entropy loss* $\ell(f, y) := -\log \sigma(f)_y$, where $\sigma : \mathbb{R}^D \to \mathbb{R}^D$ denotes the softmax function. In this computation we use the following facts:

$$f_\theta^{lin}(x_n) = f_\theta(x_n), \tag{12}$$

$$\nabla_\theta f_\theta^{lin}(x_n) := \nabla_{\theta'} f_{\theta'}^{lin}(x_n)\big|_{\theta'=\theta} = \phi(x_n) = \nabla_\theta f_\theta(x_n), \tag{13}$$

$$\nabla_\theta^2 f_\theta^{lin}(x_n) := \nabla_{\theta'}^2 f_{\theta'}^{lin}(x_n)\big|_{\theta'=\theta} = 0, \tag{14}$$

which follow directly from (10).

### B.1 SECOND-ORDER APPROXIMATION FOR SQUARED LOSS

For the squared loss, we obtain:

$$\nabla_\theta^2 \ell \left( f_\theta^{lin}(x_n), y_n \right) = \nabla_\theta^2 \left[ \frac{1}{2} (f_\theta^{lin}(x_n) - y_n)^T (f_\theta^{lin}(x_n) - y_n) \right] \tag{15}$$

$$= (f_\theta^{lin}(x_n) - y_n)^T (\nabla_\theta^2 f_\theta^{lin}(x_n)) + (\nabla_\theta f_\theta^{lin}(x_n))^T (\nabla_\theta f_\theta^{lin}(x_n)) \tag{16}$$

$$= \phi(x_n)^T \phi(x_n) \tag{17}$$

$$= \phi(x_n)^T R_n \phi(x_n), \tag{18}$$

where $R_n := 1_{D \times D}$ is here the $D \times D$-identity matrix. This is Eq. (9) for the squared loss.

Due to this diagonal form of $R_n$, the matrix multiplication $(\phi(x_n)\theta_s)^T R_n (\phi(x_n)\theta_{s'})$ in Eq. (5) has, for each pair $(s, s')$, a computational complexity *linear* in the dimension $D$ (which is the number of NN outputs), rather than quadratic:

$$(\phi(x_n)\theta_s)^T R_n (\phi(x_n)\theta_{s'}) = (\phi(x_n)^T \theta_s)(\phi(x_n)\theta_{s'}) \tag{19}$$

$$= \sum_{j=1}^{D} (\phi(x_n)_j \theta_s)_j (\phi(x_n)\theta_{s'})_j, \tag{20}$$

where $(\phi(x_n)\theta_s)_j \in \mathbb{R}$ denotes the $j$-th component of the vector $\phi(x_n)\theta_s \in \mathbb{R}^D$.

It is for this reason that the overall computational complexity of SOSP-I is linear in $D$ (see Sect. 2.1), rather than of order $O(D^2)$.

## B.2 SECOND-ORDER APPROXIMATION FOR CROSS-ENTROPY LOSS

Note that the first derivatives of the softmax-function $\sigma : \mathcal{R}^D \to \mathcal{R}^D$, defined by $\sigma(f)_i :=$ $e^{f_i} / \sum_{k=1}^D e^{f_k}$ are:

$$\frac{\partial}{\partial f_j} \sigma(f)_i = -\sigma(f)_i (\sigma(f)_j - \delta_{ij}). \tag{21}$$

We can thus compute for the cross-entropy loss, where $\delta_{y.} \in \mathbb{R}^D$ denotes the vector with entry 1 in component $y$ and entries 0 everywhere else:

$$\nabla_\theta^2 \ell \left( f_\theta^{lin}(x_n), y_n \right) = -\nabla_\theta^2 \left[ \log \sigma(f_\theta^{lin}(x_n))_{y_n} \right] \tag{22}$$

$$= \nabla_\theta \left[ (\nabla_\theta f_\theta^{lin}(x_n))^T (\sigma(f_\theta^{lin}(x_n)) - \delta_{y.}) \right] \tag{23}$$

$$= (\nabla_\theta f_\theta^{lin}(x_n))^T (\nabla_\theta \sigma(f_\theta^{lin}(x_n))) + (\nabla_\theta^2 f_\theta^{lin}(x_n))^T (\sigma(f_\theta^{lin}(x_n)) - \delta_{y.}) \tag{24}$$

$$= (\nabla_\theta f_\theta^{lin}(x_n))^T \cdot \left( \nabla_\theta \, \sigma(f_\theta^{lin}(x_n)) \right). \tag{25}$$

To compute $\nabla_\theta \sigma(f_\theta^{lin}(x_n))$ in (25), we consider its $i$-th component:

$$\nabla_\theta \, \sigma \left( f_\theta^{lin}(x_n) \right)_i = \sum_{j=1}^D \left. \frac{\partial \sigma(f)_i}{\partial f_j} \right|_{f=f_\theta^{lin}(x_n)} \cdot \nabla_\theta (f_\theta^{lin}(x_n))_j \tag{26}$$

$$= \sum_{j=1}^D \sigma(f_\theta^{lin}(x_n))_i \left( \delta_{ij} - \sigma(f_\theta^{lin}(x_n))_j \right) \cdot \nabla_\theta \left( f_\theta^{lin}(x_n) \right)_j \tag{27}$$

$$= \sum_{j=1}^D (R_n)_{ij} \cdot \nabla_\theta \left( f_\theta^{lin}(x_n) \right)_j \tag{28}$$

$$= \left( R_n \cdot \nabla_\theta \, f_\theta^{lin}(x_n) \right)_i, \tag{29}$$

where $R_n \in \mathbb{R}^{D \times D}$ is the matrix with entries

$$(R_n)_{ij} = \sigma(f_\theta^{lin}(x_n))_i \delta_{ij} - \sigma(f_\theta^{lin}(x_n))_i \, \sigma(f_\theta^{lin}(x_n))_j \tag{30}$$

$$= \sigma \left( f_\theta(x_n) \right)_i \delta_{ij} - \sigma(f_\theta(x_n))_i \, \sigma(f_\theta(x_n))_j. \tag{31}$$

Thus, plugging back into (25),

$$\nabla_\theta^2 \ell \left( f_\theta^{lin}(x_n), y_n \right) = \left( \nabla_\theta f_\theta^{lin}(x_n) \right)^T R_n \left( \nabla_\theta f_\theta^{lin}(x_n) \right) \tag{32}$$

$$= \phi(x_n)^T R_n \, \phi(x_n), \tag{33}$$

which is Eq. (9) for the cross-entropy loss.

## B.3 SECOND-ORDER APPROXIMATION FOR OTHER LOSS FUNCTIONS

The same developments as above apply to any twice differentiable loss function $\ell$: The Hessian of the loss with the linearized model $f_\theta^{lin}$ instead of the full network $f_\theta$ can be written as

$$\nabla_\theta^2 \, \ell \left( f_\theta^{lin}(x_n), y_n \right) = \phi(x_n)^T R_n \phi(x_n), \tag{34}$$

where $R_n = \nabla_z^2 \ell(z, y_n) \big|_{z=f_\theta^{lin}(x_n)} \in \mathbb{R}^{D \times D}$ has a different form depending on the specific loss function. Thus, a general loss function $\ell$ still allows the use of the approximation used by SOSP-I. The only caveat is that $R_n$ in general can be any $D \times D$-matrix (without sparsity or low-rank properties), so that matrix multiplication may in general not be as efficient as for the above loss functions (described in detail in App. B.4); in this case the second complexity term for SOSP-I in Eq. (7) can worsen from $O(N'DS^2)$ to $O(N'D^2S^2)$.

### B.4 EFFICIENT MATRIX MULTIPLICATION FOR SQUARED LOSS AND CROSS-ENTROPY LOSS

Note that $R_n$ is a sum of a diagonal matrix $R_n^{\text{diag}}$ (first part in Eq. (31)) and a rank-1 matrix $R_n^{\text{rank-1}}$ (second part in Eq. (31)). Due to this special matrix form, the matrix multiplication $(\phi(x_n)\theta_s)^T R_n(\phi(x_n)\theta_{s'})$ in Eq. (5) has, for each pair $(s, s')$, a computational complexity *linear* in the dimension $D$ (the number of NN outputs), rather than quadratic. For the diagonal part $\left(R_n^{\text{diag}}\right)_{ij} = \sigma\left(f_\theta(x_n)\right)_i \delta_{ij}$, the reason for this is similar to the one for the squared error:

$$(\phi(x_n)\theta_s)^T R_n^{\text{diag}}(\phi(x_n)\theta_{s'}) = \sum_{i,j=1}^{D} (\phi(x_n)\theta_s)_i \left(R_n^{\text{diag}}\right)_{ij} (\phi(x_n)\theta_{s'})_j \tag{35}$$

$$= \sum_{j=1}^{D} (\phi(x_n)\theta_s)_i \, \sigma\left(f_\theta(x_n)\right)_i (\phi(x_n)\theta_{s'})_i. \tag{36}$$

For the rank-1 part $\left(R_n^{\text{rank-1}}\right)_{ij} = -\sigma(f_\theta(x_n))_i \, \sigma(f_\theta(x_n))_j$, the $O(D)$-efficient computation is

$$(\phi(x_n)\theta_s)^T R_n^{\text{diag}}(\phi(x_n)\theta_{s'}) = \sum_{ij} (\phi(x_n)\theta_s)_i \left(R_n^{\text{diag}}\right)_{ij} (\phi(x_n)\theta_{s'})_j \tag{37}$$

$$= -\sum_{ij=1}^{D} (\phi(x_n)\theta_s)_i \, \sigma(f_\theta(x_n))_i \, (\phi(x_n)\theta_{s'})_j \, \sigma(f_\theta(x_n))_j \tag{38}$$

$$= -\left(\sum_{i=1}^{D}(\phi(x_n)\theta_s)_i \, \sigma(f_\theta(x_n))_i\right) \cdot \left(\sum_{j=1}^{D}(\phi(x_n)\theta_{s'})_j \, \sigma(f_\theta(x_n))_j\right). \tag{39}$$

Again, for these reasons, the overall computational complexity of SOSP-I is linear in $D$ (see Sect. 2.1), instead of quadratic in $D$. This can make a significant difference for some datasets (e.g. $D = 1000$ classes on ImageNet).

An alternative $O(D)$-efficient way of computing our second-order approximation follows by continuing from Eq. (25), noting that $\nabla_\theta f_\theta^{lin}(x_n) = \nabla_\theta f_\theta(x_n)$ by (13):

$$\nabla_\theta^2 \ell\left(f_\theta^{lin}(x_n), y_n\right) = (\nabla_\theta f_\theta(x_n))^T \cdot (\nabla_\theta \, \sigma(f_\theta(x_n))) \tag{40}$$

$$= \phi(x_n)^T \cdot \phi^\sigma(x_n), \tag{41}$$

where we defined $\phi^\sigma(x_n) := \nabla_\theta\left(\sigma(f_\theta(x_n))\right) \in \mathbb{R}^{D \times P}$. Thus, each term in the sum (5) can be written as

$$(\phi(x_n)\theta_s)^T(\phi^\sigma(x_n)\theta_{s'}) = \sum_{j=1}^{D}(\phi(x_n)\theta_s)_j \, (\phi^\sigma(x_n)\theta_{s'})_j, \tag{42}$$

which again has complexity $O(D)$. For this it is necessary to pre-compute each $\phi^\sigma(x_n)$ in addition to $\phi(x_n)$, but both have the same complexity.

We finally note that our approximation of the second-order derivatives (i.e., of the Hessian) is somewhat different from the approximation made in (Peng et al., 2019) for the cross-entropy case: While we dropped all second-order derivatives $\nabla_\theta^2 f_\theta(x_n)$ of the *pre*-softmax activations, Peng et al. (2019) dropped all second-order derivatives of the *post*-softmax activiations, i.e. all terms $\nabla_\theta^2\left(\sigma(f_\theta(x_n))\right)$. A consequence of this difference is that our approximation (33) (or (42)) does not depend on the labels $y_n$ (this is already apparent from our intermediate step Eq. (25)), whereas the approximation made in (Peng et al., 2019) does depend on the labels $y_n$. The fact that our second-order approximation is independent of the $y_n$ is similar to the second-order approximation in the squared-loss case (see App. B.1 above, and also (Peng et al., 2019)). Note furthermore that the first-order terms in the loss approximation (see e.g. in Eq. (2) or (3)) are the same in our method as in (Peng et al., 2019)), and these do depend on the labels $y_n$ (cf. the expression in square bracket in Eq. (23)).

## C  SECOND-ORDER APPROXIMATION CORRESPONDS TO OUTPUT-CORRELATION

The purpose of this section is to provide a better intuition of the second-order components of our loss approximation. First, we reformulate the expression for the second-order terms in Eq. 5. We use the fact that for any ReLU-NN without batch normalization layers it holds almost everywhere that

$$\nabla_\theta f_\theta(x)\theta_s = f_{\theta^s}(x),\qquad(43)$$

where $\theta^s$ is the weight vector, where all structures of the layer that contains structure $s$ are set to zero expect for the weights of structure $s$ itself. The identity is straightforward to derive, therefore we show the relation for a feed-forward neural network with zero biases, but the proof for a convolutional neural network is almost identical .

Let $f_\theta$ be a fully-connected neural network with $L$ layers and $f_\theta(x) = W_L \mathbf{1}_{h^{L-1}(x)\geq 0} W_{L-1}...\mathbf{1}_{h^1(x)\geq 0} W_1 x$, where $W_l$ are the weight-matrices and $h^l(x)$ the output functions of the $l$-th layer and $\mathbf{1}_{h^l(x)\geq 0}$ the diagonal matrix with the step function on the diagonal elements corresponding to the components of $h^l(x) \geq 0$. Now assuming that structure $s$ is contained in the $i$-th layer, the only non-vanishing components of the vector $\theta_s$ are the once associated with structure $s$. Thus, one can evaluate the gradient to get

$$\nabla_\theta f_\theta(x)\theta_s = W_L \mathbf{1}_{h^{L-1}(x)\geq 0} W_{L-1}...\mathbf{1}_{h^i(x)\geq 0} W_i^s...\mathbf{1}_{h^1(x)\geq 0} W_1 x,\qquad(44)$$

where $W_i^s$ is the weight matrix of layer $i$ where all components are set to zero except for those belonging to structure $s$. Using this, one directly receives $\nabla_\theta f_\theta(x)\theta_s = f_{\theta^s}(x)$

Next, applying this identity to Eq. 5 gives

$$\theta_s^T H(\theta)\theta_{s'} \approx \frac{1}{N'}\sum_{n=1}^{N'}(\phi(x_n)\theta_s)^T R_n(\phi(x_n)\theta_{s'})$$

$$= \frac{1}{N'}\sum_{n=1}^{N'}(\nabla_\theta f_\theta(x_n)\theta_s)^T R_n(\nabla_\theta f_\theta(x_n)\theta_{s'})$$

$$= \frac{1}{N'}\sum_{n=1}^{N'}f_{\theta^s}(x_n)R_n f_{\theta^{s'}}(x_n),$$

where the explicit form of the $R_n$ matrix is provided in the previous section. From this relation one can now see that the second-order components of our pruning objective correspond to output-correlations. This connection could also explain, why our second-order pruning methods do not improve the pruning performance at initialization, but do improve performance after training, since at initialization different structures may not have as strong output-correlations as the learned features after training.

## D  COUNTING OF PARAMETERS AND MACS

Here we provide details on how we compute the number of parameters of our pruned NNs, and the number of MACs required for the evaluation of a pruned NN on one input point. The description is specific to the ResNets and DenseNets used in our work; the main complication arises for the ResNet architecture (in particular for the residual connections), as we describe below.

While our parameter and MAC counting is exact, it is somewhat intricate. We do not know whether this exact counting has been implemented in other papers on pruning as well (see the comparisons in Tables 1 and 2), since these other works did not elaborate on their counting. Therefore, the comparability with the counts from other papers is not necessarily given. When we compare our exact counting to a more straightforward but approximative counting, we find that our exact counts for ResNet50 (Table 2) yield substantially higher parameter and MAC numbers than the straightforward approximate counting. The straightforward approximate counting would yield MAC-pruning-ratios that are 12-18 percentage points higher (better) than our exact numbers. We now explain our exact counting first.

The number of parameters of a convolutional layer $l$ equals the number $F_l^{in}$ of input filters of the layer times the number $F_l^{out}$ of output filters times the kernel size $K_l^{wh}$ (which equals the kernel width times the kernel height): $C^l = F_l^{in} \cdot F_l^{out} \cdot K_l^{wh}$. In addition to this count, there are $2F_l^{out}$ parameters for the batchnorm layer following each convolutional layer (our networks do not have bias terms in the convolutional layers, which would add another $F_l^{out}$).

The subtlety is now that, within the chain of convolutional layers in the ResNet architecture, the number $F_{l+1}^{in}$ of input filters into the following convolutional layer $l+1$ does *not necessarily* equal the number of output filters $F_l^{out}$ of the present convolutional layer $l$. Namely, this can happen if the output of layer $l$ is added to the output of a residual connection to compute the input into $l+1$. At such a point in the chain of convolutional layers, the number of input filters $F_{l+1}^{in}$ into layer $l+1$ depends on both the output filters of layer $l$ *as well as* on the output filters of the residual connection.

More precisely, only those filters can be removed from the input into layer $l+1$ which are absent from *both* the output of layer $l$ *as well as* from the output of the residual connection. (Note, thus, that the number $F_{l+1}^{in}$ of input filters into layer $l+1$ *cannot* be determined by only knowing the number of output filters $F_l^{out}$ and the number of output filters of the residual connection. Rather, the answer depends on *which* output filters have been pruned.) To determine $F_{l+1}^{in}$, two kinds of residual connections have to be distinguished:

(a) **Residual connection is an identity skip connection.** In this case, the output filters of the residual connection are exactly the output filters of a previous layer: either the output filters of layer $l' := l - 2$ (when the skip connection is in a "BasicBlock") or of layer $l' := l - 3$ (when the skip connection is in a "BottleneckBlock"). $F_{l+1}^{in}$ thus equals the number of filters that are un-pruned in the output of both layer $l'$ and un-pruned in the output of layer $l$.

(b) **Residual connection is a downsampling layer.** As we exclude downsampling layers from pruning (see Sec. 3), the number of output filters of a downsampling layer in the pruned network equals the number of outputs of the downsampling layer in the original network. Therefore, if the output of layer $l$ is added to the output of a downsampling layer, then $F_{l+1}^{in}$ in the pruned network takes the same value as in the original (un-pruned) network, i.e. $F_{l+1}^{in}$ is the original number of input filters into layer $l+1$.

The same reasoning and procedure applies to determining the number of input filters into any downsampling layer (i.e., this number is the same as the number of inputs into the convolutional layer that has the same input as the downsampling layer; note, a downsampling layer is a convolutional layer as well, but not part of the "chain of convolutional layers"), and also to determine the number of inputs ("input neurons") into the final fully-connected layer (i.e., the number of "input neurons" into the last fully-connected layer is the same as would be the number of input filters into a convolutional layer at this stage).

Having determined the number of input and output filters (neurons) into all channels in this way, our parameter count sums up the parameter numbers for all convolutional layers (incl. downsampling layers), batchnorm layers, and fully-connected layers (incl. bias terms). This is the exact number of parameters needed to specify the pruned NN and also to build the pruned NN, since these parameters specify all surviving filters (and the fully-connected layers).

Our exact MAC count is similar, also based on the "true" $F_l^{in}$ and $F_l^{out}$ as just determined. The MAC count of a convolutional layer is $M_l = C_l \cdot S_l^w \cdot S_l^h$, where $S_l^w$ and $S_l^h$ are the numbers of width-wise and height-wise applications of each filter; for our networks, $S_l^w$ equals the spatial picture width at layer $l$ divided by the width-stride, and similarly for $S_l^h$.

Finally – and as mentioned above – we briefly describe the more straightforward but approximate counting method, that would yield pruning ratios that can appear quite a bit better. In this approximate way of counting, we take, within the chain of convolutional layers in the ResNet architecture, as input filters into layer $l+1$ exactly the output filters from layer $l$, i.e. $F_{l+1}^{in} := F_l^{out}$. Consequently for any downsampling layer, we take as its number of input filters the number of output filters of its preceding convolutional layer, and as its number of output filters we take the number of input filters into the following convolutional layer (as determined by the previous sentence). We compute the number of parameters and MACs then according to the same formulas as in the exact method, but with potentially other values for $F_l^{in}$ and $F_l^{out}$ for all convolutional layers (incl. the downsampling

layers, and the number of input nodes into the last fully-connected layer). Note, this approximate count is actually the exact count for a version of the pruned network where the size of the residual connections (identity skip connections and downsampling layers) has been adapted in a "natural" way to the sizes of the pruned convolutional layers in the chain of convolutional layers. In particular, this count can therefore be computed by just knowing the numbers of pruned filters in each convolutional layer, instead of knowing exactly which of the filters in each convolutional layer have been pruned.

It is apparent from the way of approximate counting just described that its parameter and MAC count will be smaller or equal to the exact count (described futher above); both counts conincide for the original (un-pruned) network. For pruned networks, however, the difference in both counts can be substantial, esp. for the ResNet50 network (Table 2). For example, whereas the MAC-pruning-ratio of our SOSP-I(0.3) method is 15% (see Table 2), its MAC-pruning-ratio according to the approximate counting would equal 27%. In case that the competitor papers used this simpler approximate counting (which we do not know), we should also use this approximate counting to evaluate our method, which would thus appear more favorable, especially on the ImageNet experiments and especially on ResNet-50 (Table 2).

On a final note, we remark that the paper Tang et al. (2020) introducing the SCOP method, mentioned a discrepancy between the theoretically computed number of MACs and the experimentally measured value for this quantity, hinting at least at some inconsistency in the counting.

