# OpenReview forum: "SOSP: Efficiently Capturing Global Correlations by Second-Order Structured Pruning"
_ICLR.cc/2022/Conference — ICLR 2022 Spotlight_

### Official Review · Reviewer_q96Z · 2021-10-27

**Correctness:** 3
**Technical Novelty And Significance:** 2
**Empirical Novelty And Significance:** 3
**Recommendation:** 6
**Confidence:** 3

**Main Review:**

Strengths(+):
1. The proposed approximation schemes for second-order structured pruning is interesting with theoretical derivations and rigorous experimental test.
2. The paper is well written and the presentation is clear.

Weaknesses(-):
1. I am concerned on the advantages of using second-order information to improve the pruning performance for networks with large number of parameters, as can be seen from Fig. 4, the performance gap between first-order method and SOSP is quite small when #parameters reach 750K. Modern large-scale networks usually have even more parameters.
2. The Fig.5 also raises some concerns since the gap on test accuracy between first-order and SOSP is only around 0.1%~0.3% percent, which has quite limited improvement.
3. It seems that the merit of capturing global correlation and good scalability is not achieved by a single approximation method, but by the SOSP-I and SOSP-H respectively. This may decrease the technical attractiveness of the paper.

Minors:
1. Maybe due to space limitation, some important Figures are put into supplementary materials, the readers need to read back and forth to get a clear understanding.

**Summary Of The Paper:**

This paper develops two novel saliency-based pruning methods for second-order structured pruning which consider correlations among all structures and layers and have good scalability to large-scale neural network structures and datasets. The proposed method can also be used to find and remove architectural bottlenecks, which further improves the performance of the pruned networks.

**Summary Of The Review:**

Due to the limited potential performance improvement in practical benchmark database, my overall rating would be (weak) rejection

---

> ### Author Response · Authors · 2021-11-11
> **Response to Reviewer q96Z**
>
> We thank reviewer q96z for their questions and feedback. We address these in the following:
>
> 1. *I am concerned on the advantages of using second-order information to improve the pruning performance for networks with large number of parameters, as can be seen from Fig. 4, the performance gap between first-order method and SOSP is quite small when #parameters reach 750K. Modern large-scale networks usually have even more parameters.*
>
> For Fig. 4, we start with a pretrained vanilla ResNet-56 and then continuously reduce the parameter count by increasing the pruning ratio of both pruning procedures, respectively. The accuracy achieved after fine-tuning is then plotted over the number of effective parameters that the pruned model has. Therefore, when increasing the number of parameters, both SOSP and first-order pruning must naturally converge to the same accuracy because for a pruning ratio of 0.0 both models equal the full network and must achieve the same accuracy. We hope this addresses the reviewer’s concern. We revised the figure caption accordingly to prevent future misunderstandings and thank the reviewer for pointing this out.
>
> Further, if the reviewer’s concern is that adding the second-order terms is not beneficial for large-scale models, we kindly refer the reviewer to the ImageNet results in Tab. 2. In this comparison, SOSP significantly outperforms the results of the first-order method Taylor, which is very similar to our SOSP-H method except that it drops all second-order terms. This demonstrates that SOSP-H significantly improves the achieved accuracies by adding second-order terms while only marginally increasing the computational resources to compute the saliency.
>
> 2. *The Fig.5 also raises some concerns since the gap on test accuracy between first-order and SOSP is only around 0.1%~0.3% percent, which has quite limited improvement.*
>
> We would like to point out that Fig. 5 does not show the comparison of a first-order method to our second-order SOSP methods (this comparison is shown in Fig. 4), but actually a comparison between SOSP with and without second-order correlations (i.e., the off-diagonal terms of the Hessian). Thus, Fig. 5 shows that adding these second-order correlations can significantly improve the performance of our pruning procedure. This is a key ingredient in achieving state-of-the-art performance.
>
> If the intention of the reviewer was to point towards the comparison between SOSP and first-order pruning, we kindly refer the reviewer to the answer provided for the previous question (see 1.). Lastly, we are wondering where the reviewer’s mentioned numbers 0.1% to 0.3% come from, because SOSP can outperform its reduced variants in Figs. 4 and 5 by much larger margins.
>
> 3. *It seems that the merit of capturing global correlation and good scalability is not achieved by a single approximation method, but by the SOSP-I and SOSP-H respectively. This may decrease the technical attractiveness of the paper*
>
> We think there might be a misunderstanding regarding the attributes of SOSP-I and SOSP-H. Both algorithms capture global correlations since both algorithms make full use of the off-diagonal terms of the Hessian. Regarding scalability, SOSP-H can be conveniently applied to modern large-scale networks, since it has a lower computational complexity than SOSP-I (see Section 2.3 and the experiments). Additionally, both methods have a lower computational complexity than calculating the full second-order terms without any approximations (also see Section 2.3) .
> We have revised Section 4 in the new manuscript version to clarify that both algorithms are capturing global correlations efficiently, as also described in Section 2. We thank the reviewer for pointing this out.
>
> 4. *Maybe due to space limitation, some important Figures are put into supplementary materials, the readers need to read back and forth to get a clear understanding.*
>
> Unfortunately, due to space limitations, we could not include all our sanity checks and ablation studies into the main manuscript. We included the figures that support our main arguments in the main text and point the interested reader to specific places in the supplementary material for additional ablation studies. If the reviewer has a specific suggestion on which figure in the main text to replace by which another figure, please let us know.
>
> We hope that our answers can resolve the reviewer’s concerns and questions.

---

### Official Review · Reviewer_7bu1 · 2021-11-01

**Correctness:** 4
**Technical Novelty And Significance:** 3
**Empirical Novelty And Significance:** 3
**Recommendation:** 8
**Confidence:** 2

**Main Review:**


For the SOSP-H, it assumes the pruning ratios should be high and then the approximation would be held. In Tab.1 and Tab.2, I can see that the reduction ratios of weights for Densenet40 on Cifar10/100 are around 50% and similarly for ResNet 18/50 on ImageNet. So I suggest the authors clarify what ratios are considered to be high.

In Tab.2, when comparing with the other methods, the performance of the proposed methods is on par with the SOTA. Whether the pruning efficiency will outperform the other methods. I can see in Fig. 1 the authors demonstrated results on Cifar, but how about on ImageNet?

I am not an expert on network pruning. However, I am wondering why the compared methods are from 2019 and 2020. Whether some references published in 2021, such as Liu et al. 2021, Hayou et al. 2021, Su et al. 2021, can be compared?

In Fig. 3, the orange bars are not clear to me. The vertical axes are representing different meanings that are not easy for readers to understand. For c and f, whether the pruning mainly focuses on input and low-level layers? Why did those layers have low pruning ratios originally? Maybe the authors could explain it and visualise filters before and after pruning?

I think the comparison with CCP is also needed in Tab. 1 as it is also based on second-order correlations.

Last but not least, when I recap the formulation of this paper, from Eq. 1 to 6, I also wonder if in general eq. 6 is the main contribution of this paper while eq.5 is not scalable (though it has very nice deduction). Whether eq. 5 is not necessary for practical problems?

**Summary Of The Paper:**

This paper presents saliency-based second-order structured pruning methods, namely SOSP-I and SOSP-H. The proposed methods are designed to capture the correlations among all structures and layers, known as second-order structure (Hessian). In particular, SOSP-I employs Hessian approximation while SOSP-H employs exact Hessian. Overall, the idea of this paper is very clear, and I kinda like the discussion part of it. The expand-pruning is also interesting.

**Summary Of The Review:**

Overall, I think this paper has its merits. However, as I am not an expert in this field but a general reader. I am lean to accept this paper.

---

> ### Author Response · Authors · 2021-11-11
> **Response to Reviewer 7bu1 (Part 1)**
>
> We thank reviewer 7bu1 for the positive feedback and their comments and questions. We respond to each comment and question separately below:
>
> 1. *For the SOSP-H, it assumes the pruning ratios should be high and then the approximation would be held. In Tab.1 and Tab.2, I can see that the reduction ratios of weights for Densenet40 on Cifar10/100 are around 50% and similarly for ResNet 18/50 on ImageNet. So I suggest the authors clarify what ratios are considered to be high.*
>
> In principle, the approximation used by SOSP-H is tight for high pruning ratios, while the approximation used by SOSP-I is independent of the pruning ratio but becomes tight for models with low training loss. In practice, as we see in our experiments, the performance (accuracy) of SOSP-H is comparable to that of SOSP-I even for small pruning ratios (lower than ca. 50%). This observation supports the choice of approximations for SOSP-H and allows to apply SOSP-H with any pruning ratio.
>
> 2. *In Tab.2, when comparing with the other methods, the performance of the proposed methods is on par with the SOTA. Whether the pruning efficiency will outperform the other methods. I can see in Fig. 1 the authors demonstrated results on Cifar, but how about on ImageNet?*
>
> The results shown in Tab. 2 and in the corresponding Fig. 6 show our comparisons for ImageNet. Due to space limitations we preferred to include Tab. 2 into the main text and show its visualization in Fig. 6 in the supplementary material. Our method SOSP outperforms all other compared methods on ImageNet.
>
> 3. *I am not an expert on network pruning. However, I am wondering why the compared methods are from 2019 and 2020. Whether some references published in 2021, such as Liu et al. 2021, Hayou et al. 2021, Su et al. 2021, can be compared?*
>
> We mainly compare SOSP to other single-shot pruning methods. Su et al. uses an iterative pruning approach and is thus not easily comparable to our single-shot method. In principle, our saliency pruning method could be combined with iterative procedures, but we advocate the disentanglement of saliency approximations from other effects to provide better comparisons.
>
> Liu et al. present a single-shot method that uses a conventional Fisher approximation without considering off-diagonal terms and introduces a novel method to find and jointly prune groups of structures that depend on each other.  Such grouping is an orthogonal method to ours and could also be combined with our advanced approximations of saliencies. Since we compare our results to C-OBD, which uses the same saliency approximation as Liu et al., we did not directly compare to this work.
> Lastly, Hayou et al. focus on pruning at initialization. While we also investigate the performance of SOSP for pruning at initialization in an ablation study (see App. 7-9), we focus on pruning of pretrained networks.
>
>
>
> We searched in the recent conference proceedings for other single-shot pruning methods and to the best of our knowledge compare our SOSP approximation to all other saliency approximations used for structured single-shot pruning algorithms.
>
>
> 4. *In Fig. 3, the orange bars are not clear to me. The vertical axes are representing different meanings that are not easy for readers to understand. For c and f, whether the pruning mainly focuses on input and low-level layers? Why did those layers have low pruning ratios originally? Maybe the authors could explain it and visualize filters before and after pruning?*
>
>  We agree with the reviewer that Fig. 3 is somewhat confusing. We tried to make the best use of the limited space available. The orange or rather red curve in Fig. 3 c,d,f,g depicts the multiplier of layer width over layer index. We updated Fig. 3 to clarify to the reader which axis belongs to which curve by coloring the axis labels with the same color as the data points. Thank you for pointing this out.
>
> The blue histograms in (c,d,f,g) depict the pruning ratio over layer index after widening and retraining the network. For both ResNet and VGG, our global SOSP method results in low pruning ratios for low-level layers (blue bars are small for low layer indices in Fig. 8 d,f). Consequently, we identified these layers as architectural bottlenecks and increased their width (high value for width multiplier in red in c,f) to improve the overall network accuracy (see b, e).
>
> Does this answer your question and what do you mean by visualizing filters before and after pruning?

---

> > ### Author Response · Authors · 2021-11-11
> > **Response to Reviewer 7bu1 (Part 2)**
> >
> > 5. *I think the comparison with CCP is also needed in Tab. 1 as it is also based on second-order correlations.*
> >
> > Currently, in Tab. 1 we compare our SOSP methods to other global pruning approaches and in Fig. 1 to local pruning approaches.  Since CCP is based on a second-order approach, but is a local pruning method and thus in our current setting should be compared in Fig. 1. However, CCP uses a different MAC-count which makes it difficult for us to visualize their results in the same Figure as the other local methods. We added a comparison of SOSP-I, SOSP-H and CCP over the alternative MAC-count for ResNet-56 to the appendix (now Fig. 7).
> >
> >  6. *Last but not least, when I recap the formulation of this paper, from Eq. 1 to 6, I also wonder if in general eq. 6 is the main contribution of this paper while eq.5 is not scalable (though it has very nice deduction). Whether eq. 5 is not necessary for practical problems?*
> >
> > We agree with the statement of the reviewer that SOSP-H (eq. 6) is our main contribution, and we generally recommend the use of SOSP-H. Since SOSP-I is based on more common approximations, we use SOSP-I to validate the approximation used by SOSP-H, especially for lower pruning rates.

---

### Official Review · Reviewer_6Vrj · 2021-11-03

**Correctness:** 3
**Technical Novelty And Significance:** 3
**Empirical Novelty And Significance:** 3
**Recommendation:** 8
**Confidence:** 4

**Details Of Ethics Concerns:**

I do not have ethical concerns, since this paper is a theoretical paper about neural network model pruning.

**Main Review:**

Strengths

1. This paper makes a good contribution in presenting the idea around structured-based model pruning. The mathematical derivation is very well presented and easy to follow, and the literature review is also very well presented.

2. The experiments are also promising, and they show the superior performance of the proposed approach in terms of effectiveness in reducing the parameters while keeping the classification performance.

Weakness
1. The second-order approximation seems to be specialized for squared loss and cross-entropy loss. I am not sure if the proposed approximation can be generalized to the other losses. This limits the impact of the proposed approach, as modern neural networks might involve many other forms of losses.e.g. MSELoss, NLLLoss, CTCLoss, PoissonNLLLoss. How can one adopt the proposed approximation for the wide range of loss functions?

2. Some key implementation details are missing out in the paper.

a) The fast hessian-vector product seems to be the critical component of the proposed SOSP-H method. However, it is not clear how this is exactly implemented. It is not clear how the fast hessian-vector product is implemented to achieve the second-order approximation.
Using the hessian seems to be a great idea, but I am unsure how the proposed approach addresses the storage problems. For example, storing the hessian matrix for a standard ResNet50 would take at least 2.5 Petabytes of memory. For larger-scale neural networks, how would the proposed approach be able to address them? What are the theoretical limits on the proposed method that the SOSP-H cannot handle? Given that the SOSP-I results are not as great as those with SOSP-H, I think audiences might be more interested in the SOSP-H. But the high space cost seems to be a problem.

b) The presentation arrangement of this paper could be improved significantly. Some technical terms are presented without explanation. For example, what does it mean “individual sensitives”?

3. Some comparisons are missing out. How does the proposed approximation approach compare against the empirical Fisher approximation? I only get to know more about the mathematical details of this after reading the supplementary, it would be great to add some key approximation equations to the main paper.

4. The main evaluation metrics used in this work are MAC numbers and the model Top-1 accuracy. It would be interesting to see the speed of the pruned networks on the standard devices, though this is missing in the current paper.

**Summary Of The Paper:**

This paper address the problem of neural network single-shot structured-based model pruning.

Deep convolutional neural networks grow to achieve higher performance, which also means slower inference and higher computational cost. Model pruning can help with that. Model pruning can be unstructured, which means to remove individual weights, or structured, which means to remove entire substructures, e.g., nodes or channels. This paper focuses on structured pruning.

A key challenge for global saliency-based structured model pruning is to find a good objective that can be efficiently calculated to make the approach scalable to various modern convolutional neural networks. Existing saliency-based pruning methods such as OBD, C-OBD evaluate the effect of removing a single weight or structure on the loss of the neural network in isolation. This work is to devise a 2nd order pruning method that considers all global correlations for structured sensitivity pruning.

The basic idea in this work is to search for the pruning mask M to minimize the joint effect on the network loss approximately. The mathematical approximation focuses on the second-order approximation to the loss.

The key contributions of this work are the proposed 2nd-order structured pruning (SOSP). There are two variants of that. The first one is based on fast hessian-vector products, and it has the same complexity as that with first-order methods. A second one is based on the Gaussian-newton approximation. The first one with fast hessian-vector products do better in terms of scale.

Experiments on VGG, ResNets, PlainNet, DenseNet are shown promising results across various image classification datasets, including Cifar10/100 and ImageNet.


**Summary Of The Review:**

1. This paper is an exciting work, and it would have a solid impact on the structured-based model pruning.

2. This submission also has a lot of minor issues. For example, the second-order approximation seems to be developed towards addressing particular two losses used in classification. It is not clear if the proposed approach would work for the other losses.

3. Some technical terms are not well explained.

4. The evaluation metrics used in this work should also include speed, but it is missing in the current submission.

---

> ### Author Response · Authors · 2021-11-11
> **Response to Reviewer 6Vrj (Part 1)**
>
> We thank reviewer 6Vrj for the detailed comments, questions and for the positive feedback. We will discuss the comments and questions separately below:
>
> 1. *The second-order approximation seems to be specialized for squared loss and cross-entropy loss. I am not sure if the proposed approximation can be generalized to the other losses. This limits the impact of the proposed approach, as modern neural networks might involve many other forms of losses.e.g. MSELoss, NLLLoss, CTCLoss, PoissonNLLLoss. How can one adopt the proposed approximation for the wide range of loss functions?*
>
> We thank the reviewer for bringing this to our attention. We mainly discuss the Cross-Entropy loss and the MSE loss in our work. For these two, we describe in detail how they work within SOSP-I. But in fact, SOSP-H does not need to be adapted to any specific loss function (the same formulas and the same code apply), and thus SOSP-H can be applied to any loss-function with the exact same scaling. For SOSP-I, the only change is that one would need to determine the second-order-derivative matrix R_n for each loss separately (see new Appendix B.3).  In summary, SOSP-H can be easily applied to any loss functions by drop-in replacements while SOSP-I requires a bit more specific adaptations, but the overall procedure works as described for the other loss functions.
>
> Furthermore, this generic applicability of SOSP-H is not only an advantage over SOSP-I but also over all structured pruning schemes based on either the Fisher-approximation or any other loss-specific approximation. We think that discussing additional loss functions could be a valuable extension of our paper and have added a short discussion to App. B.3. Thank you again for pointing out this point.
>
>  2. *Some key implementation details are missing out in the paper.*
>
> - *a) The fast hessian-vector product seems to be the critical component of the proposed SOSP-H method. However, it is not clear how this is exactly implemented. It is not clear how the fast hessian-vector product is implemented to achieve the second-order approximation. Using the hessian seems to be a great idea, but I am unsure how the proposed approach addresses the storage problems. For example, storing the hessian matrix for a standard ResNet50 would take at least 2.5 Petabytes of memory. For larger-scale neural networks, how would the proposed approach be able to address them? What are the theoretical limits on the proposed method that the SOSP-H cannot handle? Given that the SOSP-I results are not as great as those with SOSP-H, I think audiences might be more interested in the SOSP-H. But the high space cost seems to be a problem.*
>
> For the Hessian-Vector product we make use of the implementation available within Pytorch. This implementation uses a simple trick which can be utilized within the autograd framework. Using this trick, the Hessian-Vector product can be calculated via two backpropagations, and thus without excessive memory consumption. Assume that we want to calculate the Hessian-Vector product of a vector v (fixed and given) with the Hessian of a function f, then the following steps are required:
>
> - Calculate the gradient of f with respect to the weights -> grad_f
>
> - Calculate the scalar product grad_f@v -> prod
>
> - Calculate the gradient of prod with respect to the weights -> grad_prod
>
> The resulting gradient of prod is equivalent to the Hessian-Vector product of H with v. This trick is also used for other applications and is implemented in Pytorch (see https://pytorch.org/docs/stable/_modules/torch/autograd/functional.html#hvp) and  Tensorflow (see https://github.com/tensorflow/tensorflow/blob/master/tensorflow/python/ops/gradients_impl.py#L924 lines 319-369).
>
> Since SOSP-H uses this efficient implementation of the Hessian-Vector product, there is no need to calculate or store the full Hessian. Thus, SOSP-H can scale to extremely large networks. Memory is not a major concern for SOSP-H because the Hessian does not need to be constructed. The required memory capacity only scales linearly with the number of parameters, as in ordinary backpropagation, rather than quadratically.
>
> - b) *The presentation arrangement of this paper could be improved significantly. Some technical terms are presented without explanation. For example, what does it mean “individual sensitives”?*
>
> We thank the reviewer for pointing this out and have tried to better explain these terms in our revised version of the paper. For the term “individual sensitivities”, which we use somewhat loosely, see the beginning of Sect. 2.1.

---

> > ### Author Response · Authors · 2021-11-11
> > **Response to Reviewer 6Vrj (Part 2)**
> >
> > 3. *Some comparisons are missing out. How does the proposed approximation approach compare against the empirical Fisher approximation? I only get to know more about the mathematical details of this after reading the supplementary, it would be great to add some key approximation equations to the main paper.*
> >
> > We agree with the reviewer that some of our derivations and additional experiments had to be put into the appendix due to space limitations. We attempted to fit Eq. (9) together with a small additional paragraph into the main text of our paper (as the reviewer mentioned, this would add some more details to the main paper), but we were not able to realize this within the page limit of 9 pages. Including all the formulas necessary for a proper comparison with the Fisher approximation into the main text would be even more extensive and thus cannot be done, also because we want to keep a balance between empirical and theoretical results in the main part of our paper. Our compromise is to have frequent pointers to the individual sections in the Appendix containing the full details. We hope this solution is acceptable to the reviewer.
> >
> > 4. *The main evaluation metrics used in this work are MAC numbers and the model Top-1 accuracy. It would be interesting to see the speed of the pruned networks on the standard devices, though this is missing in the current paper.*
> >
> > We agree with the reviewer that speed of the pruned networks could be an interesting comparison but conducting a fair comparison can be challenging because speed varies across different hardware systems, e.g. different GPUs, and even for identical hardware similar utilization has to be guaranteed across all experiments. Consequently, we used the number of MACs and effective parameters to evaluate the computational cost of pruned networks, which is also usually a quite accurate estimation for embedded devices.

---

> > > ### Comment · Reviewer_6Vrj · 2021-11-25
> > > **Response to the authors**
> > >
> > > Thanks for addressing the comments. I am happy to recommend accepting the paper.
> > >
> > > But I still think it is interesting to mention the speed of the proposed approach with an off-the-shelf GPU device so that people would have at least a rough idea of the proposed approach is practically useful.

---

### Official Review · Reviewer_3yXC · 2021-11-08

**Correctness:** 3
**Technical Novelty And Significance:** 3
**Empirical Novelty And Significance:** 3
**Recommendation:** 8
**Confidence:** 3

**Details Of Ethics Concerns:**

None.

**Main Review:**

On the strengths

1. Network pruning is important for deep learning approach and its wide application to practical tasks.
2. It is sound to consider the second-order correlation information among the structures of neural networks to conduct pruning.
3. Experimental study demonstrates the overall better performance of the proposed SOSP-H with respect to the relevant methods proposed in the literature;
4. This papers provides detailed discussion and supplementary materials to explain the proposed work.

On the weaknesses
1. Some parts on proposing SOSP can be made clearer. For example, it mentions Eq. (2) as "a modification of Eq.(1)." However, the relationship between Eq.(1)  and Eq.(2) is not clearly discussed. It seems that Eq.(2) is a (much) relaxed upper bound of Eq.(1)? In this case, how accuracy can Eq.(2) approximate Eq.(1)? The similar issue applies to the relationship between Eq. (6) to Eq.(1). Please clarify.
2. The logic behind deciding the architectural bottleneck could be made more precise. Say, if some layers are barely pruned, does this actually mean they are so important that they shall not be removed? In this case, how to arrive at the idea in this work that "removing these could further improve the performance of the model?” Please clarify.
3. This paper provides detailed discussion in Section 4, which is appreciated. Meanwhile, the summary of this discussion could be put before  Section 2 as Related Work. This will help to introduce the existing literature on second-order information based pruning and highlight the differences of the proposed work from the existing ones. This helps to better capture the novelty of this work.


**Summary Of The Paper:**

This paper proposes a method to prune deep neural networks. The aim is to reduce the computational cost and inference time while maximally maintaining the classification performance. The motivation of this work is to consider second-order structured pruning (SOSP), which considers the correlation information among the structures and layers when conducting network pruning. The key part of this work is the development of a method called SOSP-H that can have better scalability while considering the second-order correlation information for pruning. Experimental study is conducted to compare the proposed SOSP-H with its variant and other existing related methods, demonstrating its effectiveness.

**Summary Of The Review:**

This is a piece of work of good quality, with detailed information. A rigorous and principled approach is taken by this work to address the second-order pruning issue. Experimental study demonstrates the overall competitive performance of the proposed method. At the same time, some parts of this work can be further clarified to make the argument more precise and clearer. The novelty and contribution with respect to the existing relevant methods can be better described. Also, the structure of the presentation can be improved a bit.

---

> ### Author Response · Authors · 2021-11-11
> **Response to Reviewer 3yXC**
>
> We thank reviewer 3yXC for the positive feedback and discuss the specific comments and questions raised by the reviewer in detail below:
>
> 1. *Some parts on proposing SOSP can be made clearer. For example, it mentions Eq. (2) as "a modification of Eq.(1)." However, the relationship between Eq.(1) and Eq.(2) is not clearly discussed. It seems that Eq.(2) is a (much) relaxed upper bound of Eq.(1)? In this case, how accuracy can Eq.(2) approximate Eq.(1)? The similar issue applies to the relationship between Eq. (6) to Eq.(1). Please clarify.*
>
> It is correct that Eqs. (2) and (6) are relaxed upper bounds on (1). However, the primary motivation for (2) and (6) is not to provide a bound on (1), but rather to prevent cancellations between the individual sensitivity terms, resulting in a better assessment of the importance of structures. Furthermore, experimentally the saliency measure with absolute values around individual sensitivities in (2) and (6) outperforms other possible saliency measures in which individual sensitivities can cancel each other out as in (1). We hope this clarifies the transition from (1) to (2) and (6). We tried to make this clearer in our improved manuscript.
>
> 2. *The logic behind deciding the architectural bottleneck could be made more precise. Say, if some layers are barely pruned, does this actually mean they are so important that they shall not be removed? In this case, how to arrive at the idea in this work that "removing these could further improve the performance of the model?” Please clarify.*
>
> We thank the reviewer for this question. Our previously used expression “removing architectural bottlenecks” might indeed be misleading here. In the expand-pruning scheme we do _not_ remove the layer(s) that we identify as an architectural bottleneck, but we rather _widen_ the respective layer(s), such that the bottleneck goes away. We suspect that this clarified terminology already answers the reviewer’s comment.
>
> To elaborate on our underlying idea, the rationale behind the detection of architectural bottlenecks is that we try to identify those layers within our network with the lowest pruning rate. Therefore, the individual structures within these “bottleneck layers” contribute a disproportionately large amount to the overall accuracy or loss. Thus, we conclude that widening these layers could reduce the impact of each individual structure on the overall loss/accuracy, therefore making the bottleneck disappear. This allows us to further improve in the performance of our pruned models (see Fig. 3) by creating smaller models with an overall lower drop in performance, compared to the baseline model.
> We agree with the reviewer that the term “removing” might have been misleading here, and we substituted the term “removing a bottleneck” by “widening a bottleneck”. We hope this resolves the reviewer’s concern.
>
> 3. *This paper provides detailed discussion in Section 4, which is appreciated. Meanwhile, the summary of this discussion could be put before Section 2 as Related Work. This will help to introduce the existing literature on second-order information based pruning and highlight the differences of the proposed work from the existing ones. This helps to better capture the novelty of this work.*
>
> As acknowledged by the reviewer, a detailed discussion of our work and of related work can be found in Section 4. We found it sensible to discuss the related work in the light of our results, after our results section, and thus in Section 4. Due to space limitations, we are unable to add a longer summary of the related work discussion from Section 4 before Section 2, but we added a reference there to point the reader to the “related work/discussion” section in Section 4. We thank the reviewer for this comment.

---

### Author Response · Authors · 2021-11-11
**Revision of the paper PDF**

We thank all reviewers for the valuable feedback we have received. Since we are continuously updating our manuscript to incorporate the changes and additions suggested by the reviewers, we want to list here the changes to the manuscript already incorporated:

1. Added comments to make the intuition behind the transition from Eq. (1) to (2) and (6) clearer.

2. Substituted the wording “removing bottlenecks” with “widening bottlenecks”.

3. Added before Section 2 a reference to the “related work” discussion in Section 4 to point readers there.

4. Added subsection B.3 to Appendix B, where we discuss how SOSP can be applied to other loss functions beyond Cross-Entropy and MSE loss, and added corresponding comments to Sect. 2.

5. Added further explanations to some technical terms (e.g., individual sensitivities)

6. Updated Fig. 3 to clarify which axis belongs to which curve (red and blue coloring).

7. Added the comparison of SOSP-I, SOSP-H and CCP over the alternative MACs for ResNet-56 to the appendix (Fig. 7 in, App. A.7)

8. Improved the descriptions of Figs. 4 and 5.

---

### Decision · Program_Chairs · 2022-01-20

**Decision:**

Accept (Spotlight)

**Comment:**

Four reviewers have evaluated this submission with one score 6 and three scores 8. Overall, reviewers like the work and note that *a rigorous and principled approach is taken by this work*. AC agrees and advocates an accept.